# A GAME THEORETIC APPROACH TO META-LEARNING NASH MODEL-AGNOSTIC META-LEARNING

## ABSTRACT

Meta-learning, or learning to learn, aims to develop algorithms that can quickly adapt to new tasks and environments. Model-agnostic meta-learning (MAML), proposed as a bi-level optimization problem, is widely used as a baseline for gradient-based meta-learning algorithms that learn meta-parameters. In MAML, task-specific parameters are adapted independently in the inner-loop. After learning the task-specific parameters, the meta-parameters are learned in the outer-loop by minimizing the average task loss. After MAML, some gradient-based meta-learning research has explored objectives beyond average task losses, such as minimizing worst-case task losses for risk management and improving zero-shot performance in unadaptable environments. However, if the purpose of learning meta-parameters changes, the inner-loop formulation must change accordingly. Therefore, we propose a novel gradient-based meta-learning framework that imposes joint strategy sets and utility functions among tasks, making each task affected by other tasks. To solve this complex problem, we first show the proposed framework can be formulated as a generalized Stackelberg game. After that, we propose the NashMAML algorithm to compute the generalized Stackelberg equilibrium of this model and theoretically prove its convergence. We validate our approach on sinusoidal regression and few-shot image classification tasks. The results demonstrate that our approach outperforms previous methods in handling few-shot learning problems.

## 1 INTRODUCTION

Meta-learning, also known as learning to learn (Thrun & Pratt, 1998), aims to develop algorithms that enable more efficient adaptation to new unseen tasks, but similar to previous tasks, by learning from a variety of tasks. To achieve this goal, meta-learning algorithms are trained on a set of related tasks or domains to learn a more general set of skills (Nam et al., 2022) or priors (Finn et al., 2018; Kim et al., 2018) that can be applied to new tasks with limited data. Among them, model-agnostic meta-learning (MAML) (Finn et al., 2017) is a gradient-based meta-learning algorithm that can be applied to various different problems.

After the emergence of MAML, numerous follow-up studies have been conducted within the machine learning community (Nichol et al., 2018; Zintgraf et al., 2019; Rajeswaran et al., 2019). These studies formulate meta-learning as a bi-level optimization problem and find an optimal solution via learning task-specific parameters independently in the inner-loop (lower level problem) first, then learning meta-parameters in the outer-loop (upper level problem) to minimize the average loss of the tasks after adaptation. However, optimally minimizing the individual task losses in the inner-loop may not essentially lead to minimizing the average loss in the outer-loop. Furthermore, if the goal of the outer-loop changes, the current inner-loop formulation, which adapts the model to individual tasks independently, does not help learn the meta-parameter. For instance, the purpose of learning meta-parameters can involve minimizing the worst-case loss (Collins et al., 2020) for risk management, enhancing zero-shot performance (Nooralahzadeh et al., 2020) in unadaptable environments, or increasing training stability.

To address these limitations, we propose a new algorithm, Nash model-agnostic meta-learning (NashMAML), which was inspired by the Nash equilibrium of a game, that enables alignment of the learning objectives between the inner-loop and outer-loop. We formulate the NashMAML by

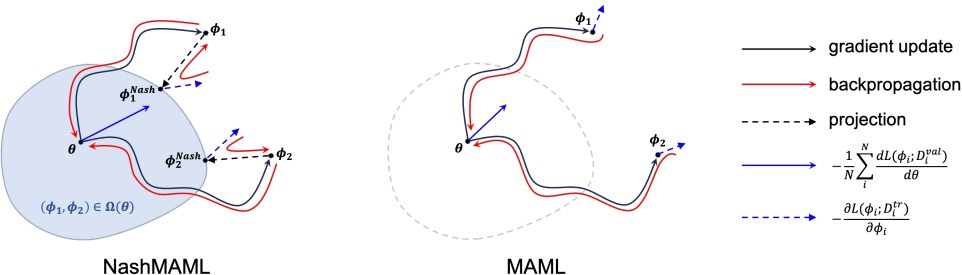

Figure 1: The training process of NashMAML and MAML. The key feature of NashMAML is the presence of feasible regions (blue region) of task parameters as determined by the joint strategy sets. The task-specific parameters $\phi_1, \phi_2$ are projected to feasible regions whenever they are located outside the feasible regions during the inner-loop training.

adding joint strategy sets and utility functions, both of which introduce the dependency among tasks to the inner-loop. Depending on the form of the joint strategy sets and joint utility functions, the NashMAML can be learned for various purposes. On the contrary, as shown in Figure 1, the conventional approach of independently optimizing task-specific parameters is no longer available for solving the inner-loop problem due to the influence of other task-specific parameters on each task.

To compute the joint optimal task-specific parameters, we adopt a game-theoretic interpretation, wherein a batch of $N$ tasks in the inner-loop is regarded as decision-makers who determine its task-specific parameters. To be specific, we model the inne-loop problem of NashMAML as a generalized Nash game for $N$ tasks, whose solution is a generalized Nash equilibrium of the tasks. Furthermore, we consider the meta-learner as a decision maker who determines meta-parameters before the task-specific parameters are determined. Then, the bi-level interactions among the meta-learner and $N$ tasks can be formulated as a generalized Stackelberg game designed to model the interaction among the leader and the followers (Stackelberg et al., 1952). The solution of the generalized Stackelberg game is a generalized Stackelberg equilibrium.

Our main contributions to this paper are as follows:

- We interpret the current meta-learning algorithms' formulation and solution concept from a game-theoretical perspective.
- We propose a novel bi-level formulation of gradient-based meta-learning as a generalized Stackelberg game. This formulation enables alignment of the learning objectives between the inner-loop and outer-loop by incorporating joint strategy sets and utility functions between tasks.
- We propose a NashMAML algorithm, which can compute the equilibrium of the proposed generalized Stackelberg game. We provide conditions for the convergence of NashMAML algorithm and its convergence speed.
- We introduce a practical example of NashMAML by proposing a ball-shaped strategy set and joint penalty function, suppressing task-specific parameters from moving away from meta parameters. For the ball-shaped strategy set, we propose a methodology in which computing the gradient with backpropagation is tractable.
- We demonstrate the proposed formulation's and NashMAML algorithm's effectiveness by conducting a comparative analysis on sinusoidal regression and image classification tasks. The results provide a potential for our approach to enhance performance, particularly in problems with complex task distributions.

## 2 RELATED WORKS

### 2.1 COMPUTATIONAL APPROACHES OF MODEL-AGNOSTIC META-LEARNING

After the proposal of MAML, various follow-up studies have been introduced to address the challenges of MAML, mostly focusing on few-shot image classification tasks. Implicit MAML Ra-

jeswaran et al. (2019) proposed Hessian-free methods, providing computational advantages compared to explicit differentiation methods. The FOMAML Finn et al. (2017) and Reptile Nichol et al. (2018) explore different strategies to approximate the outer-loop gradient update in MAML using first-order approximation approaches, effectively reducing the memory and time complexity while preserving performance. These studies share the objective, consistent with MAML, of minimizing the average task loss.

## 2.2 VARIATIONS IN OBJECTIVES OF META-LEARNING

In addition to minimizing the average loss, various studies have been conducted with other objectives. Task-Robust MAML (Collins et al., 2020) and TaRo-BOBA (Gu et al., 2021) aim to improve the worst-case performance by minimizing the maximum task loss. (Kim et al., 2018) introduces Bayesian frameworks and designed a new meta-learning objective with chaser loss to effectively model the uncertainty during the meta-learning process. In addition, instead of finding meta-parameters that perform well after adaptation, (Nooralahzadeh et al., 2020; Verma et al., 2020) focus on enhancing zero-shot performance.

## 3 PRELIMINARIES

### 3.1 GAME THEORY

Game theory is the discipline that models scenarios where multiple decision-makers aim to optimize their respective objectives. A game consists of players who make decisions, their feasible regions (or strategies), and their objective functions (or utilities). Depending on the representation methods and information structures, there are various types of games. First, we discuss the $N$ player (generalized) Nash game (Nash Jr, 1950) in which $N$ players make decisions simultaneously.

**Definition 1** *Let $G = \left\langle \mathbf{P}, (u_i)_{i \in \mathbf{P}}, (\Omega_i)_{i \in \mathbf{P}} \right\rangle$ be a $N$ players' generalized Nash game which is formulated as*

$$\max_{\mathbf{x}_i \in \Omega_i(\mathbf{x}_{-i})} u_i(\mathbf{x}_i, \mathbf{x}_{-i}), \forall i \in \mathbf{P} \tag{1}$$

*where $\mathbf{P} = \{1, \cdots, N\}$ is a set of players and $u_i$ is the utility function of the player $i$, $\mathbf{x}_i$ is the player $i$'s decision belonging to their strategy set $\Omega_i(\mathbf{x}_{-i})$, $\mathbf{x}_{-i} = (\mathbf{x}_1, \cdots, \mathbf{x}_{i-1}, \mathbf{x}_{i+1}, \cdots, \mathbf{x}_N)$ is the player's joint decision except player $i$. Then, we refer to $\mathbf{x}^* \in \prod_{i \in \mathbf{P}} \Omega_i(\mathbf{x}^*_{-i})$ as a generalized Nash equilibrium of the $N$ player's generalized Nash game $G$ if it satisfies the following equation.*

$$\mathbf{x}_i^* = \arg \max_{\mathbf{x}_i \in \Omega_i(\mathbf{x}^*_{-i})} u_i(\mathbf{x}_i, \mathbf{x}^*_{-i}), \forall i \in \mathbf{P} \tag{2}$$

*Let $\mathbf{S}(M)$ be a randomly selected $M$ player which is a subset of $N$ players, and $\mathbf{x}_{-i}(M) = (\mathbf{x}_j)_{j \in \mathbf{S}(M)-\{i\}}$ be the players' joint decision except player $i$. Then, $\mathbf{x}^* \in \prod_{i \in \mathbf{P}} \Omega_i$ is a generalized $M$-subNash equilibrium if it satisfies the following equation for every $\mathbf{S}(M)$.*

$$\mathbf{x}_i^* = \arg \max_{\mathbf{x}_i \in \Omega_i(\mathbf{x}^*_{-i}(M))} u_i(\mathbf{x}_i, \mathbf{x}^*_{-i}(M)), \forall i \in \mathbf{S}(M) \tag{3}$$

*Let $\mathbf{x}^{\text{VE}} \in \prod_{i \in \mathbf{P}} \Omega_i(\mathbf{x}^{\text{VE}}_{-i})$ be a variational equilibrium of $G$ if it satisfies the following variational inequality.*

$$\left( \frac{\partial u_i(\mathbf{x}^{\text{VE}})}{\partial \mathbf{x}_i} \right)_{i \in \mathbf{P}}^{\text{T}} (\mathbf{x}^{\text{VE}} - \mathbf{x}) \geq 0, \forall \mathbf{x} \in \prod_{i \in \mathbf{P}} \Omega_i(\mathbf{x}_{-i}) \tag{4}$$

*When the player $i$'s strategy set is independent of the other players' decisions, we refer to $G$ as a $N$ player's Nash game. The Nash equilibrium, subNash equilibrium, and variational equilibrium of the Nash game are defined in the same way as the equilibrium of the generalized Nash game described in equations (2) - (4).*

Next, we discuss the $1 - N$ Stackelberg game, where a leader makes decisions first, and then $N$ followers make decisions simultaneously after observing the leader's decision.

**Definition 2** *Let* $\Gamma = \left\langle \{1\}, \mathbf{F}, u^{\mathrm{L}}, (u_i)_{i \in \mathbf{F}}, \Omega^{\mathrm{L}}, (\Omega_i)_{i \in \mathbf{F}} \right\rangle$ *be a* $1 - N$ *generalized Stackelberg game where* $\mathbf{F} = [N]$ *is a follower set,* $u^{\mathrm{L}}$ *is an utility function of a leader,* $u_i$ *is an utility function of follower* $i$, $\Omega^{\mathrm{L}}$ *is a strategy set of a leader,* $\Omega_i$ *is a strategy set of follower* $i$. *When the follower* $i$'s *strategy set is independent of the other followers' decisions, we refer to* $\Gamma$ *as a* $1 - N$ *Stackelberg game. Then,* $(\mathbf{y}^*, \mathbf{x}^*) \in \Omega^{\mathrm{L}} \times \prod_{i \in \mathbf{F}} \Omega_i \left( \mathbf{y}^*, \mathbf{x}^*_{-i} \right)$ *is a optimal solution if it satisfies the following equation.*

$$\sup_{\mathbf{x}^*(\mathbf{y}^*) \in \mathbf{S}(\mathbf{y})} u^{\mathrm{L}} \left( \mathbf{y}^*, \mathbf{x}^* \left( \mathbf{y}^* \right) \right) \geq \sup_{\mathbf{x}^*(\mathbf{y}) \in \mathbf{S}(\mathbf{y})} u^{\mathrm{L}} \left( \mathbf{y}, \mathbf{x}^* \left( \mathbf{y} \right) \right), \forall \mathbf{y} \in \Omega^{\mathrm{L}} \tag{5}$$

*where* $S(\mathbf{y})$ *is a generalized Nash equilibrium of the* $N$ *followers' (generalized) Nash game given leader's decision* $\mathbf{y}$. *In detail,* $(\mathbf{y}^*, \mathbf{x}^*) \in \Omega^{\mathrm{L}} \times \prod_{i \in \mathbf{F}} \Omega_i \left( \mathbf{y}^*, \mathbf{x}^*_{-i} \right)$ *is a (generalized) Stackelberg equilibrium if* $S$ *is a set of (generalized) Nash equilibrium of followers, a (generalized)* $M$-*subStackelberg equilibrium if* $S$ *is a set of (generalized)* $M$-*subNash equilibrium, and a variational Stackelberg equilibrium if* $S$ *is a set of variational equilibrium of followers.*

### 3.2 GAME THEORETICAL INTERPRETATION OF MAML

The meta-learning problem is generally modeled as bi-level programming ($1 - 1$ Stackelberg game) since tasks are independent of each other. The purpose of solving task $i$ is to learn task-specific parameters $\phi_i$ using a dataset $\mathcal{D}_i^{\mathrm{tr}}$ to minimize the loss function $\mathcal{L} \left( \phi_i; \mathcal{D}_i^{\mathrm{tr}} \right)$. Then, the meta-parameters $\theta$ aim to minimize the average loss across all the tasks.

The problem that model-agnostic meta-learning (MAML) and first-order MAML (FOMAML) algorithms intend to solve is defined as the bi-level programming as follows.

$$\theta^* = \arg \min_{\theta \in \mathbb{R}^d} F(\theta) := \frac{1}{N} \sum_{i=1}^{N} \mathcal{L} \left( \phi_i^*(\theta); \mathcal{D}_i^{\mathrm{val}} \right) \tag{6}$$

$$\phi_i^*(\theta) = \arg \min_{\phi_i \in \mathbb{R}^d} \mathcal{L} \left( \phi_i; \mathcal{D}_i^{\mathrm{tr}} \right), \forall i \in [N] := \{1, \cdots, N\} \tag{7}$$

The MAML (Finn et al., 2017) and FOMAML (Nichol et al., 2018) formulation reflect that task-specific parameter vector $\phi_i$ is close to the meta-parameter vector $\theta$ not through the problem structure but by controlling the number of inner steps. They approximately compute $\phi_i^* \sim \hat{\phi}_i = \theta - \alpha \frac{\partial \mathcal{L} \left( \theta; \mathcal{D}_i^{\mathrm{tr}} \right)}{\partial \phi_i}$ through the finite number of gradient updates. However, since $\hat{\phi}_i$ is different from the optimal task-specific parameters $\phi_i^*(\theta)$ as defined in equation (7), the resulting meta-parameters is not the optimal solution (Stackelberg equilibrium) of the bi-level programming in equations (6)-(7).

The MAML and FOMAML algorithms have limitations in that they control the proximity of task-specific parameters to the meta-parameters through the number of inner steps rather than the problem formulation. It means that the meta-parameters computed by adjusting the number of inner steps are not the optimal meta-parameters $\theta^*$ as defined in equation (6). Therefore, MAML and FOMAML algorithms have poorer performance than other meta-learning algorithms that compute the optimal meta-parameters.

The game theoretic interpretations for implicit MAML (iMAML) and fast context adaptation via meta-learning (CAVIA) algorithms, which are famous extensions of MAML, are discussed in Appendix A.

## 4 NASH MODEL-AGNOSTIC META-LEARNING

MAML has a fixed formulation of the lower level problem designed to improve task adaptation performance. However, if the purpose of learning meta-parameters at the upper level changes, it is necessary to change the lower level formulation accordingly. For instance, the objective of learning meta-parameters may extend beyond merely minimizing the average task loss. It can involve minimizing the worst-case loss for risk management, enhancing zero-shot performance in unadaptable environments, or increasing training stability. To address this issue, we present a novel bi-level

formulation and algorithm that considers the mutual interaction among tasks of the same batch by applying joint strategy sets or utility functions based on the formulation of MAML. Note that, this framework can be generally adapted to other gradient-based meta-learning algorithms, such as iMAML and CAVIA.

## 4.1 FORMULATION

First, we formulate the target problem of NashMALM from a stochastic optimization perspective. We denote the joint task-specific parameter $(\phi_i)_i$ as $\phi$, and $N$ randomly sampling joint task-specific parameter except parameter $i$, $(\phi_j)_{j \in \mathbf{S}(N)-i}$, as $\phi_{-i}$ where $\mathbf{S}(N) = \{i | \mathcal{T}_i \sim p(\mathcal{T})\}$ is the set of $N$ randomly sampling tasks' index. The target problem of the NashMAML algorithm where the batch size is $N$ is the following stochastic bi-level problem and its optimal solution is $(\theta^*, \phi^*(\theta^*))$.

$$\theta^* = \arg \min_{\theta \in \mathbb{R}^d} \mathbb{E}_{\mathcal{T}_i \sim p(\mathcal{T})} [\mathcal{L}_i(\theta, \phi_i^*(\theta))] \tag{8}$$

$$\phi_i^*(\theta) = \arg \min_{\phi_i \in \Omega_i(\phi_{-i}^*(\theta), \theta)} f_i(\phi_i, \phi_{-i}^*(\theta), \theta) \tag{9}$$

where task $i$'s strategy set $\Omega_i(\phi_{-i}^*(\theta), \theta)$ depends on the other task-specific parameters $\phi_{-i}$, and task $i$'s utility function $f_i(\phi_i, \phi_{-i}, \theta) = \mathcal{L}_i(\phi_i) + g(\phi_i, \phi_{-i}, \theta)$ is the sum of task $i$'s loss function $\mathcal{L}_i$ and $g$, the function affected by the $\phi_{-i}$ and $\theta$. This formulation makes the lower level problem target other purposes, other than task loss, imposed by $\Omega_i$ and $g$.

However, in practice, learning meta-parameters is conducted in batch units, and the problems addressed during a single meta-parameters update can be precisely formulated. We model the single meta-parameter's gradient update of the NashMAML algorithm as a $1 - N$ generalized Stackelberg game $\Gamma = \left\langle \{1\}, [N], F, (f_i)_{i \in [N]}, \mathbb{R}^d, (\Omega_i)_{i \in [N]} \right\rangle$ where leader's decision is a meta-parameter $\theta$, and followers' decision are their respective task-specific parameter $\phi_i$. The set of the follower is $[N] = \{1, \cdots, N\}$ where $N$ is batch size, leader's utility function is $F(\theta, \phi) = \frac{1}{N} \sum_{i=1}^N \mathcal{L}_i(\phi_i; \mathcal{D}_i^{\mathrm{val}})$, follower $i$'s utility function is $f_i(\phi_i, \phi_{-i}, \theta) = \mathcal{L}_i(\phi_i; \mathcal{D}_i^{\mathrm{tr}}) + g(\phi_i, \phi_{-i}, \theta)$, leader's strategy set is $\mathbb{R}^d$, and follower $i$'s strategy set is $\Omega_i$. Then, the optimal solution $(\theta, \phi^*(\theta)) \in \mathbb{R}^d \times \Omega(\theta)$ of $\Gamma$ satisfies the following equations (10) and (11) where $\Omega(\theta) = \prod_{i \in [N]} \Omega_i(\theta, \phi_{-i}^*(\theta))$ is a generalized Stackelberg equilibrium.

$$\theta^* = \arg \min_{\theta \in \mathbb{R}^d} F(\theta, \phi^*(\theta)) \tag{10}$$

$$\phi_i^*(\theta) = \arg \min_{\phi_i \in \Omega_i(\phi_{-i}^*(\theta), \theta)} f_i(\phi_i, \phi_{-i}^*(\theta), \theta), \forall i \in [N] \tag{11}$$

where $\phi = (\phi_i)_{i \in [N]}$ is a joint task-specific parameter and $\phi_{-i} = (\phi_1, \cdots, \phi_{i-1}, \phi_{i+1}, \cdots, \phi_N)$ is a joint task-specific parameter except task $i \in [N]$.

## 4.2 ALGORITHM

In the NashMAML algorithm, we first compute the generalized Nash equilibrium $\phi^*(\theta) = (\phi_i^*(\theta))_{i \in [N]}$ of the lower level problem as defined in equation (11). Next, we explicitly compute $\frac{d\phi^*(\theta)}{d\theta}$ through back-propagation of $\phi^*(\theta)$ to obtain the optimal meta-parameters $\theta^*$. Thus, the solution computed by the NashMAML algorithm is a generalized $N$-subStackelberg equilibrium of the generalized Stackelberg game as formulated by equations (8)-(9). We describe the NashMAML algorithm, which is an extension of MAML, in the Algorithm 1, detailed in Appendix B. The difference between NashMAML and MAML is that including a projection step onto the strategy set if the joint task-specific parameter is not feasible for the strategy set.

## 4.3 THEORETICAL RESULT

First, we define the following estimator to measure the error of the estimated gradient.

Table 1: Complexity for the meta-learning algorithms

| Algorithm | Iteration complexity | Memory |
|---|---|---|
| MAML (GD, full back-prop) | $\kappa \log (D/\delta)$ | $\mathrm{Mem}\,(\nabla \mathcal{L}_i)\,\kappa \log (D/\delta)$ |
| MAML (Nesterov's AGD, full back-prop) | $\sqrt{\kappa} \log (D/\delta)$ | $\mathrm{Mem}\,(\nabla \mathcal{L}_i)\,\sqrt{\kappa} \log (D/\delta)$ |
| implicit MAML (Nesterov's AGD) | $\sqrt{\kappa} \log (D/\delta)$ | $\mathrm{Mem}\,(\nabla \mathcal{L}_i)$ |
| NashMAML (PRGD, full back-prop) | $\kappa \log (D/\delta)$ | $\mathrm{Mem}\,(\nabla \mathcal{L}_i)\,\kappa \log (D/\delta)$ |

**Definition 3** *Let the joint task-specific parameter $\hat{\phi}$ be a solution estimated by a computing algorithm (e.g. PRGD method). Then, $\hat{\phi}$ is a $\delta$-accurate estimation of the optimal joint task-specific parameter $\phi^*$ if it satisfies the following:*

$$\left\| \hat{\phi} - \phi^* \right\| \leq \delta \tag{12}$$

**Definition 4** *Let $\frac{\hat{d}F}{d\theta}$ be an approximated gradient of the meta loss function. Then, $\hat{h}_\theta$ is an $\epsilon$-accurate estimation of the meta loss function if it satisfies the following:*

$$\left\| \frac{\hat{d}}{d\theta} F\left(\theta, \hat{\phi}\right) - \hat{h}_\theta\left(\theta, \hat{\phi}\right) \right\| \leq \epsilon \tag{13}$$

Table 1 summarizes the iteration complexity to compute $\frac{d\phi^*(\theta)}{d\theta}$ of NashMAML and the conventional meta-learning algorithms, MAML and iMAML. Importantly, the iteration complexity to compute $\frac{d\hat{\phi}_i(\theta)}{d\theta}$ of the NashMAML is equivalent to the conventional algorithms as $O\left(\log(D/\delta)\right)$ from the perspective of error, $\delta$. Moreover, the memory complexity of the NashMAML is equivalent to the conventional algorithm as $O\left(\mathrm{Mem}\,(\nabla \mathcal{L}_i)\,\kappa \log (D/\delta)\right)$ where $\mathrm{Mem}\,(\nabla \mathcal{L}_i)$ is the memory taken to compute a single derivative $\nabla \mathcal{L}_i$ (Rajeswaran et al., 2019). We discuss the complexity to compute $\frac{d\hat{\phi}_i(\theta)}{d\theta}$ in the following theorem.

**Theorem 1** *Let $D$ be the diameter of search space of the joint task-specific parameter $\phi = (\phi_i)_{i \in [N]}$ in the inner optimization problem (i.e. $\|\phi - \phi^*(\theta)\| \leq D$). Suppose that the projected reflected gradient descent (PRGD) method (Malitsky, 2015) is used to compute the $\delta$-accurate estimation of the optimal joint task-specific parameter $\hat{\phi} = \left(\hat{\phi}_i\right)_{i \in [N]}$ of the generalized Nash equilibrium, which is the convergent point of the inner-loop of the NashMAML algorithm. Under Assumption 1, the NashMAML algorithm computes $\hat{\phi}$ with $O\left(\kappa \log (D/\delta)\right)$ number of iterations, and only $O\left(\mathrm{Mem}\,(\nabla \mathcal{L}_i)\,\kappa \log (D/\delta)\right)$ memory is required throughout.*

The remaining part covers the algorithm for finding the equilibrium of the lower level which holds not only for the PRGD method but also for general cases. The second main result is that we compute the error of the estimated gradient $\hat{h}_\theta$ through back-propagation is bounded by a weighted sum of the error in estimating task-specific parameters $\phi$ and the error in estimating gradient through back-propagation.

**Theorem 2** *Let $\theta$ be a given meta-parameter, $\phi^*$ be an optimal task-specific parameter, $\hat{\phi}$ be a $\delta$-accurate estimated task-specific parameter, and $\hat{h}_\theta$ be an $\epsilon$-accurate estimated gradient of $F$ with respect to $\theta$ computed through back-propagation. Under Assumption 2, the difference between the $\epsilon$-accurate estimated gradient $\hat{h}_\theta$ and the gradient of the optimal meta loss function $F$ with respect to $\theta$, $\frac{dF}{d\theta}$, is bounded by the weighted sum of the error in estimating $\phi^*$ and the error in estimating the gradient through back-propagation. That is,*

$$\left\| \frac{d}{d\theta} F\left(\theta, \phi^*(\theta)\right) - \hat{h}_\theta\left(\theta, \hat{\phi}\right) \right\| \leq C \left\| \phi^*(\theta) - \hat{\phi} \right\| + \left\| \frac{\hat{d}}{d\theta} F\left(\theta, \hat{\phi}\right) - \hat{h}_\theta\left(\theta, \hat{\phi}\right) \right\|$$

$$\leq C\delta + \epsilon \tag{14}$$

*where $C = L_1 + \frac{C_1 L_4 + C_2 L_2}{\mu_1 + \mu_2} + \frac{C_1 C_2 (L_3 + L_5)}{(\mu_1 + \mu_2)^2}$.*

Now, we prove the convergence of the NashMAML algorithm in Theorem 3, and prove the convergent point of the NashMAML algorithm is the generalized subStackelberg equilibrium of the stochastic optimization problem described in equations (8) and (9) in Theorem 4.

**Theorem 3** *Let $(\theta^*, \phi^*(\theta^*))$ be a convergent point of the NashMAML algorithm, and $\mathcal{L}(\theta)$ be an expected optimal meta loss function, that is, $\mathcal{L}(\theta) = \mathbb{E}_{\mathcal{T}_i \sim p(\mathcal{T})}[F(\theta, \phi^*(\theta))]$. Under Assumptions 1, 2, and 3, the following statements hold.*

- *The expected difference of the meta-parameter $\mathbb{E}_{\mathcal{T}_i \sim p(\mathcal{T})}\left[\left\|\theta^{k+1} - \theta^k\right\|^2\right]$ is bounded.*

- *The expected difference of the optimal meta loss function $\mathcal{L}(\theta^{k+1}) - \mathcal{L}(\theta^k)$ is bounded.*

- *The expected error of the optimal meta loss function of the convergent point $\mathcal{L}(\theta^*) - \mathcal{L}(\theta^k)$ is bounded.*

**Theorem 4** *Let $(\theta^*, \phi^*(\theta^*))$ be an optimal solution of the stochastic optimization problem described in equations (8) and (9) which is the target problem of the NashMAML algorithm. We denote the expected meta loss function of the stochastic optimization problem $\mathbb{E}_{\mathcal{T}_i \sim p(\mathcal{T})}[\mathcal{L}_i(\theta, \phi^*(\theta))]$ as $\mathbb{E}[\mathcal{L}_i^*(\theta)]$. Let $\delta$ and $\bar{\delta}$ are the convergence criterion of the inner-loop and the outer-loop, respectively. Then, under Assumptions 1, 2, and 3, the NashMAML algorithm with step size $\beta \leq \frac{\bar{\delta}}{\sqrt{4C^2\delta^2 + 4\left(\bar{C}_1 + \frac{C_1 C_2}{\mu_1 + \mu_2}\right)^2}}$ compute the optimal solution of the stochastic optimization problem described in equations (8) and (9) with the convergence speed $O\left(\max\left\{k_b^2, k_\sigma^2, k_{\bar{\sigma}}^2\right\}\right)$ and error*

$$\mathbb{E}\left[\mathcal{L}_i^*(\theta^*)\right] - \mathbb{E}\left[\mathcal{L}_i^*(\theta^k)\right] \leq \frac{L_6}{2}\bar{\delta}^2 + \frac{C^2\delta^2 - \left(\bar{C}_1 + \frac{C_1 C_2}{\mu_1 + \mu_2}\right)^2}{\sqrt{C^2\delta^2 + \left(\bar{C}_1 + \frac{C_1 C_2}{\mu_1 + \mu_2}\right)^2}}\bar{\delta} \tag{15}$$

*where $C = L_1 + \frac{C_1 L_4 + C_2 L_2}{\mu_1 + \mu_2} + \frac{C_1 C_2 (L_3 + L_5)}{(\mu_1 + \mu_2)^2}$.*

Finally, we prove the convergent point of the NashMAML algorithm is always equivalent regardless of the order of the gradient update and the initial meta-parameter and task-specific parameters in Theorem 5.

**Theorem 5** *Under Assumption 1, 2, and 3, the NashMAML algorithm converges to the same optimal solution of the stochastic optimization problem described in equations (8) and (9) regardless of the order of the task-specific parameters' gradient update in the inner-loop. Moreover, the NashMAML algorithm converges to the optimal solution of the stochastic optimization problem described in equations (8) and (9) regardless of the initial meta-parameter and initial task-specific parameters under Assumption 4.*

All proofs and details for the theorems are in Appendix C.2.

Overall, we prove that the NashMAML algorithm consistently converges to the same point, regardless of the meta-parameters (initial parameter of task-specific parameters) or the order of task-specific parameters updates within the same batch. We also show that this convergent point corresponds to the generalized N-subStackelberg equilibrium of the equation (8)-(9)..

## 5 EXPERIMENTS

In this section, we propose two formulations for lower level problems to leverage optimal solutions without overfitting individual tasks while balancing the impact of tasks on the training of meta-parameters. After that, we present numerical experiments to demonstrate the potential of our framework. Our primary focus is to validate the theoretical findings and provide empirical evidence of the effectiveness of our proposed approach. For this purpose, we have chosen the 1D sine regression task as a representative example of a simple problem that clearly illustrates the core concepts and benefits of our game-theoretical framework. To show the scalability of our practical algorithm, we conduct experiments on few-shot image classification tasks.

## 5.1 PRACTICAL FORMULATION OF NASHMAML

We conduct experiments with two different formulations of lower level problems. First, we formulate the NashMAML with the penalty function by regularizing the distance between the meta-parameter and the average of task-specific parameters as follows:

$$\phi_i^*\left(\theta\right) = \arg\min \mathcal{L}_i\left(\phi_i\right) + \frac{\lambda}{2}\left\|\theta - \frac{\phi_i + \sum_{j\neq i}\phi_j}{N}\right\|_2 \tag{16}$$

Second, we formulate the NashMAML with the joint strategy set by assigning a ball-shaped constraint that limits the sum of distances between the meta-parameter and each task-specific parameter as follows

$$\phi_i^*\left(\theta\right) = \arg\min_{\phi_i \in \Omega_i\left(\phi_{-i}^*(\theta),\theta\right)} \mathcal{L}_i\left(\phi_i\right) \tag{17}$$

where the joint feasible strategy set of the task $i$ with hyperparameter $r \in \mathbb{R}_+$ is given by

$$\Omega_i\left(\theta, \phi_{-i}\right) = \left\{\phi_i \in \mathbb{R}^d \;\middle|\; \|\phi_i - \theta\|_2^2 + \sum_{j\neq i}\|\phi_j - \theta\|_2^2 \leq r^2\right\}. \tag{18}$$

The joint utility function and strategy set in equations (16) and (18) ensure that task-specific parameters are influenced by meta-parameter and other task-specific parameters, thus preventing overfitting. Specifically, we prove that backpropagation is possible through Algorithm 1 by establishing the differentiability of the projection of task-specific parameters onto equation (18) in Appendix B.3.

## 5.2 SINUSOID REGRESSION

We consider 1D sine regression task, where each task instance $\mathcal{T}_i$ is a regression problem $y = a_i \sin(x - b_i)$. Each task is the inference of the amplitude and phase from the sampled data. For each task, the learner is given $K$ samples where each sample $x_i$ is uniformly sampled from $[-5.0, 5.0]$ and tries to approximate the underlying function in terms of mean squared error (MSE). While amplitude and phase are typically sampled from a uniform distribution, we experiment with settings where both the training and test distribution for amplitude are skewed. In particular, we sample the amplitude from $[0.1, 1.05] \cup [4.95, 5.0]$. In this setting, naively minimizing the average loss of each task without considering other tasks may lead to an unstability of training since the task distribution has two separate modes.

First, we investigate the stability of training of our method. To compute training stability, we utilize the standard deviation of training loss over the course of training. As shown in 5.2 (left), Nash-MAML with constraint shows a consistently lower standard deviation of training loss.

Figure 5.2 presents a comprehensive comparison of the test mean MSE for all evaluated models. As depicted in Figure 5.2 (middle), both versions of NashMAML consistently surpass MAML in the sine regression task. We also thoroughly examine the properties of NashMAML concerning the hyperparameter, radius ($r$). As illustrated in Figure 5.2 (right), NashMAML exhibits a high degree of robustness with respect to variations in $r$.

## 5.3 IMAGE CLASSIFICATION

We evaluate our method on a popular few-shot image classification task, the Mini-ImageNet dataset. It consists of 60,000 color images of size $84\times84\times3$, 100 classes with 600 examples per class. We use the split proposed by (Ravi & Larochelle, 2017), 64 for training, 16 for validation, and 20 for test classes. Our objective is to solve the $N$-way $K$-shot classification problem, set up as follows: Given $N$ classes, we have access to $K$ different instances of each of the $N$ classes and evaluate the performance of the model to classify new instances from the $N$ classes.

We compare our framework with FOMAML, MAML, and iMAML. While these methods are not state-of-the-art on this benchmark, they can provide an apples-to-apples comparison for studying game-theoretical analysis of gradient-based meta-learning. We also anticipate that our perspective can be extended to state-of-the-art meta-learning models with minor modifications. For a fair comparison, we use the identical convolutional architecture with baselines and follow the same training

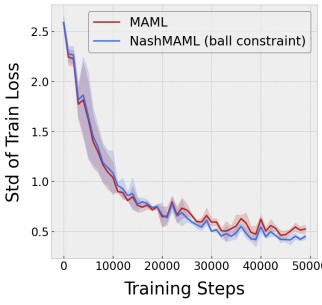 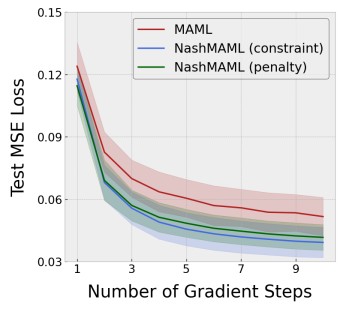 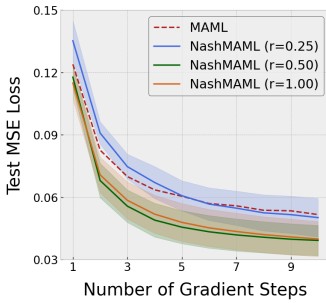

Figure 2: **1D sine regression task results.** Experiment is conducted with 3 different random seeds, mean and 95% confidence interval is reported.

procedure. To be specific, both models are trained with 5 inner gradient steps with a learning rate of 0.01 and evaluated using 10 gradient steps at test time.

Table 5.3 shows the experiment results on the Mini-ImageNet dataset. As shown in the table, Nash-MAML outperforms baselines in both 1-shot and 5-shot cases. While FONashMAML does slightly worse on the 1-shot task, it still outperforms baselines in the 5-shot task, whose performance is close to NashMAML. It seems that our formulation, which makes task-specific parameters in the inner-loop does not diverge excessively from the meta-parameters, and the generalized Stackelberg equilibrium computed by the NashMAML algorithm effectively boosts the performance of meta-learning algorithms in scalable settings.

Table 2: Mini-ImageNet 5-way $K$-shot results. FOMAML, MAML, and CAVIA results are taken from the original works (Nichol & Schulman, 2018; Zintgraf et al., 2019).

|  | 5-way 1-shot | 5-way 5-shot |
|---|---|---|
| MAML | $48.70 \pm 1.84$ % | $63.11 \pm 0.92$ % |
| NashMAML (Constraint) | $\mathbf{51.70 \pm 0.99}$ % | $\mathbf{65.34 \pm 0.65}$ % |
| NashMAML (Penalty) | $48.81 \pm 0.97$% | $62.95 \pm 0.58$% |
| FOMAML | $48.07 \pm 1.75$ % | $63.15 \pm 0.91$ % |
| FONashMAML (Constraint) | $46.70 \pm 0.99$ % | $64.12 \pm 0.23$ % |
| FONashMAML (Penalty) | $46.81 \pm 1.01$ % | $63.05 \pm 0.43$ % |
| CAVIA (32) | $47.24 \pm 0.65$ % | $59.05 \pm 0.54$ % |
| NashCAVIA (Constraint) | $46.63 \pm 0.91$ % | $59.48 \pm 0.81$ % |
| NashCAVIA (Penalty) | $47.05 \pm 0.93$ % | $60.27 \pm 0.73$ % |

# 6 CONCLUSION

In this paper, we propose a novel algorithm called NashMAML, an extension of MAML. The Nash-MAML algorithm introduces a new methodology for aligning the objective functions at the lower level to accommodate various objectives that meta-learning problems may have, such as worst-case and zero-shot performance. By assigning appropriate joint strategy sets and utility functions to the lower level based on the given upper level objectives, NashMAML ensures that the upper and lower levels share the same objectives, enabling effective learning for arbitrary objectives. In practice, we present a formulation of NashMAML focused on enhancing the stability of training and validate it through experiments in both sinusoidal regression and image classification tasks. In future research, we plan to broaden the scope beyond zero-shot performance and explore a range of objectives, including worst-case performance.

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

## A    GAME THEORETIC INTERPRETATION OF EXTENSION OF MAML

### A.1    iMAML

The implicit MAML (iMAML) (Rajeswaran et al., 2019) algorithm enforces the task-specific parameters to be close to the meta-parameters through the following bi-level problem (1-1 Stackelberg game):

$$\theta^* = \arg \min_{\theta \in \mathbb{R}^d} F(\theta) := \frac{1}{N} \sum_{i=1}^{N} \mathcal{L} \left( \phi_i^*(\theta); \mathcal{D}_i^{\mathrm{val}} \right) \tag{19}$$

$$\phi_i^*(\theta) = \arg \min_{\phi_i \in \mathbb{R}^d} \mathcal{L} \left( \phi_i; \mathcal{D}_i^{\mathrm{tr}} \right) + \frac{\lambda}{2} \|\phi_i - \theta\|^2, \forall i \in [N] \tag{20}$$

iMAML algorithm enforces the proximity of the task-specific parameters to the meta-parameters through a penalty term $\frac{\lambda}{2}\|\phi_i - \theta\|^2$. Given meta-parameters $\theta$, the iMAML algorithm computes the optimal task-specific parameters $\phi_i^*(\theta)$ of the equation (20). Then, the meta-parameters' gradient $\frac{dF(\theta)}{d\theta}$ is computed as $\sum_{i=1}^{N} \frac{\partial \mathcal{L}(\phi_i; \mathcal{D}_i^{\mathrm{val}})}{\partial \phi_i} \times \frac{d\phi_i^*(\theta)}{d\theta}$, while the iMAML algorithm computes $\frac{d\phi_i^*(\theta)}{d\theta}$ implicitly. The meta-parameters and task-specific parameters computed through the iMAML algorithm are the optimal solutions (Stackelberg equilibrium) of the bi-level programming in equations (19)-(20). The iMAML addresses the proximity of the task-specific parameter vector to the meta-parameter vector in problem formulation, resulting in improved performance compared to MAML and FOMAML (Rajeswaran et al., 2019).

### A.2    CAVIA

The CAVIA (Zintgraf et al. (2019)) algorithm divides the model parameters into two partitions, meta-parameter $\theta \in \mathbb{R}^d$ and context parameter $\phi \in \mathbb{R}^c$. In training, CAVIA learns $\theta$ at the upper level and $\phi$ at the lower level. Unlike MAML, CAVIA separates the model parameters learned at the upper and lower levels. When modeling this as a bi-level problem, it can be described as follows:

$$\theta^* = \arg \min_{\theta \in \mathbb{R}^d} F(\theta) := \frac{1}{N} \sum_{i=1}^{N} \mathcal{L} \left( \theta, \phi_i^*(\theta); \mathcal{D}_i^{\mathrm{val}} \right) \tag{21}$$

$$\phi_i^*(\theta) = \arg \min_{\phi_i \in \mathbb{R}^c} \mathcal{L} \left( \theta, \phi_i; \mathcal{D}_i^{\mathrm{tr}} \right), \forall i \in [N] \tag{22}$$

The CAVIA algorithm only uses additional context parameter $\phi$ to adapt to individual tasks at the lower level. Instead, the original model parameter $\theta$, which is shared across the tasks, is trained at the upper level. The meta-parameters and task-specific parameters computed through the CAVIA algorithm are the optimal solutions (Stackelberg equilibrium) of the bi-level programming in equations (21)-(22)

## B    EXPERIMENT DETAILS

For implementing baselines, we try to follow existing implementations from authors. For FOMAML and MAML, we use a PyTorch implementation available on github[1]. For CAVIA, we use a PyTorch implementation available on github[2]. We slightly modified the MAML and CAVIA codebase to implement our models. For all models, we perform experiments with Intel(R) Xeon(R) Gold 5317 CPU @ 3.00GHz and a single NVIDIA GeForce RTX 3090 GPU.

### B.1    SINUSOIDAL REGRESSION

We follow typical training and evaluation procedure of sinusoid regression task. We set both inner learning rate and meta learning rate to 1e-3 for all models. We use 2-layer multi-layer perceptron (MLP) with 40 hidden units and ReLU activation. For all experiments, we train meta parameter 50,000 epochs with batch size of 25.

---

[1]https://github.com/dragen1860/MAML-Pytorch
[2]https://github.com/lmzintgraf/cavia

### B.2 IMAGE CLASSIFICATION

For MiniImagenet few-shot image classification tasks, we follow the training and evaluation procedure of MAML. We use a meta batch size of 4 and 2 tasks for both models for 1-shot and 5-shot training, respectively. We use SGD optimizer with learning rate 1e-2 as an inner optimizer and Adam optimizer with learning rate 1e-3 as a meta optimizer. We train models with 60,000 iterations. For evaluation, we sample 1,000 $N$-way $K$-shot classification task instances from test dataset and compute the mean accuracy. For the CAVIA, we used the same network model described in the original paper with 32 filters and 100 context parameters.

While we follow the original hyperparameter setting, we have an additional hyperparameter, radius $r$, and regularizing coefficient $\lambda$. We do hyperparameter tuning by searching optimal $r$ in $\{1.0, 2.0, 5.0, 10.0\}$ and $\lambda$ in $\{0.1, 1.0, 5.0, 10.0\}$, and report the best performance.

### B.3 PROJECTION OF TASK-SPECIFIC PARAMETERS ON BALL-SHAPED STRATEGY SET

**Theorem 6** *Let the strategy set of task $i$ for the joint task-specific parameter $\phi = (\phi_1, \cdots, \phi_N) \in \mathbb{R}^{N \times d}$ is given by $\Omega_i(\theta, \phi_{-i}) = \{\phi_i \in \mathbb{R}^d \mid \|\phi_i - \theta\|_2^2 + \sum_{j \neq i} \|\phi_j - \theta\|_2^2 \leq r^2\}$ for the*

*fixed meta-parameter $\theta$. That is, the joint strategy set is $\Omega(\theta) = \prod_{i=1}^{N} \Omega_i(\theta, \phi_{-i}(\theta)) =$*

$\left\{\phi \in \mathbb{R}^{N \times d} \middle| \sum_{i=1}^{N} \|\phi_i - \theta\|_2^2 \leq r^2 \right\}$. *Suppose $(\phi_1, \cdots, \phi_N) \notin \Omega(\theta)$, then the projected joint task-specific parameter $\left(\phi_1^{\mathrm{Nash}}, \cdots, \phi_N^{\mathrm{Nash}}\right) \in \mathbb{R}^{N \times d}$ of $(\phi_1, \cdots, \phi_N)$ onto $\Omega(\theta)$ can be computed exactly as follow:*

$$\phi_i^{Nash} = \frac{r \times \phi_i + (d - r) \times \theta}{d} \, for \, i = 1, \cdots, N \tag{23}$$

*where $d = \left(\sum_{i=1}^{N} \|\phi_i - \theta\|_2^2\right)^{\frac{1}{2}}$.*

*Proof.* Note that the joint strategy set $\Omega(\theta)$ limits the square sum of distances from meta-parameter $\theta$ to each task-specific parameter $\phi_i$. Specifically, $d^2 = \sum_{i=1}^{N} \|\phi_i - \theta\|_2^2$ is a square sum of distance between to points $\phi_i \in \mathbb{R}^d$ and $\theta \in \mathbb{R}^d$ for $i = 1, \cdots, N$. However, it also a square of the distance between the two points $\mathbf{A} = (\theta, \cdots, \theta)$ and $\mathbf{B} = (\phi_1, \cdots, \phi_N)$ in $\mathbb{R}^{N \times d}$. Since $\mathbf{A}$ is a fixed for the given $\theta$, the joint constraint in $\mathbb{R}^{N \times d}$ represents $N \times d - \mathrm{ball}$ where the center is $\mathbf{A}$ and the radius is $r$.

According to the property of a ball, projecting a point $\mathbf{B}$ outside the ball onto the surface of the ball $\Omega(\theta)$ can be done by finding the intersection point of the line segment connecting the point $\mathbf{B}$ and the center of the ball $\mathbf{A}$ with the surface of the ball. Given that the distance between points $\mathbf{A}$ and $\mathbf{B}$ is $d$, and the radius of the ball is $r$, the projection point can be obtained by applying the formula for finding the internal division point of two points, which is shown as equation (23).

Since only multiplication, division, or root operation is used, this projection operation is differentiable, which means that it is possible to backpropagate the gradient.

## C THEOREM PROOFS

### C.1 GAME THEORETICAL ANALYSIS OF THE NASHMAML ALGORITHM

In Algorithm 1, the converged joint task-specific parameter $\phi$ computed through projected reflected gradient descent (PRGD) method is an estimated joint task-specific parameter of the optimal joint task-specific parameter $\phi^*(\theta)$. The estimated gradient of meta loss function $\hat{h}_\theta(\theta^k, \phi)$, which is computed through back-propagation, is an approximation of $\frac{d}{d\theta} F(\theta^k, \phi)$.

The PRGD method, for the inner loop of NashMAML, effectively computes the variational equilibrium of the lower-level problem. However, the lower-level problem of this algorithm addressed in

---

**Algorithm 1** NashMAML

---

**Require:** $p(\mathcal{T})$: distribution over tasks
**Require:** $\alpha, \beta$: step size hyperparameters
  randomly initialize meta-parameters $\theta^0$
    **while** $\left\| \theta^{k+1} - \theta^k \right\| \leq \bar{\delta}$ **do**
        initialize the set of task-specific parameters $\phi_i \leftarrow \theta^k$ for $i = 1, \cdots, N$
        sample batch of tasks $\mathcal{T}_i \sim p(\mathcal{T})$
        **while** $\|\phi^* - \phi\| \leq \delta$ **do**
            **for all** $\mathcal{T}_i$ **do**
                $\phi_i \leftarrow \phi_i - \alpha \frac{\partial}{\partial \phi_i} f_i\left(\phi_i, \phi_{-i}, \theta^k\right)$
            **end for**
            **if** $(\phi_1, \cdots, \phi_N) \notin \Omega(\theta)$ **then**
                $(\phi_1, \cdots, \phi_N) \leftarrow \mathrm{proj}_{\Omega(\theta)}(\phi_1, \cdots, \phi_N)$
            **end if**
        **end while**
        Update $\theta^{k+1} \leftarrow \theta^k - \beta \hat{h}_\theta\left(\theta^k, \boldsymbol{\phi}\right)$
    **end while**

---

this paper is modeled as an $N$-player generalized normal-form game, and the optimal solution is a generalized Nash equilibrium. Here, we discuss the relation between the generalized Nash equilibrium and variational equilibrium. First, we make the following assumption in order to discuss the relationship between variational equilibrium and generalized Nash equilibrium.

**Assumption 1** *Suppose the following holds:*

- *Task $i$'s strategy set of $\Omega_i$ for all tasks, and the joint strategy set $\prod_{i=1}^N \Omega_i$ are closed and convex.*

- *Task $i$'s loss function $\frac{\partial f_i}{\partial \phi_i}$ is L-smooth for all task $i$, i.e.*

$$\left\| \frac{\partial}{\partial \phi_i} f_i\left(\phi_i^1, \phi_{-i}, \theta\right) - \frac{\partial}{\partial \phi_i} f_i\left(\phi_i^2, \phi_{-i}, \theta\right) \right\| \leq L \left\| \phi_i^1 - \phi_i^2 \right\|, \forall \phi_i^1, \phi_i^2 \qquad (24)$$

- *Task $i$'s loss function $f_i\left(\phi_i, \phi_{-i}, \theta\right)$ is strongly convex for all task $i$. It means that $f_i\left(\phi_i, \phi_{-i}, \theta\right)$ is convex, and the partial gradient of loss function $\frac{\partial}{\partial \phi_i} f_i\left(\phi_i, \phi_{-i}, \theta\right)$ is $\mu$-strongly monotone for all task $i$ with condition number $\kappa = L/\mu$. $\mu$-strongly monotonicity of the gradient of the task $i$'s loss function is represented as*

$$\left\langle \frac{\partial}{\partial \phi_i} f_i\left(\phi_i^1, \phi_{-i}, \theta\right) - \frac{\partial}{\partial \phi_i} f_i\left(\phi_i^2, \phi_{-i}, \theta\right), \phi_i^1 - \phi_i^2 \right\rangle \geq \mu \left\| \phi_i^1 - \phi_i^2 \right\|^2, \forall \phi_i^1, \phi_i^2 \quad (25)$$

Now, we discuss the existence and the uniqueness of the variational equilibrium for the lower-level problem for the inner loop of the NashMAML algorithm, which is also a generalized Nash equilibrium, and prove that the PRGD method always converges to a generalized Nash equilibrium under the Assumption 1.

**Lemma 1** *Let $G(\theta) = \left\langle [N], \left(f_i\left(\phi_i, \phi_{-i}, \theta\right)\right)_{i \in [N]}, \left(\Omega_i\left(\theta, \phi_{-i}\right)\right)_{i \in [N]} \right\rangle$ be a lower-level problem of the $1 - N$ generalized Stackelberg game $\Gamma = \left\langle \{1\}, [N], F, \left(f_i\right)_{i \in [N]}, \mathbb{R}^d, \left(\Omega_i\right)_{i \in [N]} \right\rangle$ modeling the one-step meta-parameter's gradient update of the NashMAML algorithm, when meta-parameter is $\theta$. Then, there is the unique variational equilibrium of $G(\theta)$, and it is also a generalized Nash equilibrium of $G(\theta)$.*

*Proof.* By Assumption 1, $G(\theta)$ has the unique variational equilibrium by Theorem 2.3.3 of (Facchinei & Pang, 2003), and it is also a generalized Nash equilibrium by Theorem 5 of (Facchinei & Kanzow, 2010).

**Lemma 2** *Let $G\left(\theta\right) = \left\langle [N], \left(f_i\left(\phi_i, \phi_{-i}, \theta\right)\right)_{i \in [N]}, \left(\Omega_i\left(\theta, \phi_{-i}\right)\right)_{i \in [N]} \right\rangle$ be a lower-level problem of the $1 - N$ generalized Stackelberg game $\Gamma = \left\langle \{1\}, [N], F, \left(f_i\right)_{i \in [N]}, \mathbb{R}^d, \left(\Omega_i\right)_{i \in [N]} \right\rangle$ modeling the one-step meta-parameter's gradient update of the NashMAML algorithm, when meta-parameter is $\theta$. Then, the PRGD method always converges to a generalized Nash equilibrium of $G\left(\theta\right)$ under Assumption 1.*

*Proof.* By Theorem 3.3 of (Malitsky, 2015), the PRGD method always converges to the unique variational equilibrium of $G\left(\theta\right)$. By Lemma 1, the variational equilibrium computed by the PRGD method is also a generalized Nash equilibrium of $G\left(\theta\right)$.

We prove the NashMAML algorithm always converges to the generalized $N$-subStackelberg equilibrium $\left(\theta^*, \phi^*\left(\theta^*\right)\right)$ of the stochastic Stackelberg game described in equations (8) and (9) in Theorem 4. Note that the FONashMAML algorithm converges to a generalized $N$-subNash equilibrium since a meta-parameter $\theta$ is updated with partial gradients like other task-specific parameters.

## C.2 PROOF OF THEOREMS

We compute the implicit gradient for the generalized Stackelberg equilibrium of the $1 - N$ generalized Stackelberg game $\Gamma = \left\langle \{1\}, [N], F, \left(f_i\right)_{i \in [N]}, \mathbb{R}^d, \left(\Omega_i\right)_{i \in [N]} \right\rangle$ by transforming it into the $1 - 1 - 1$ Stackelberg game $\hat{\Gamma} = \left\langle f^1, f^2, f^3, \Omega^1, \Omega^2, \Omega^3 \right\rangle$ (Jo et al., 2023).

**Lemma 3** *Let $\Gamma = \left\langle \{1\}, [N], F, \left(f_i\right)_{i \in [N]}, \mathbb{R}^d, \left(\Omega_i\right)_{i \in [N]} \right\rangle$ be a $1 - N$ generalized Stackelberg game modeling the one-step gradient update of meta-parameter for the NashMAML algorithm. Then, the implicit gradient $\frac{d\phi^*(\theta)}{d\theta}$ is computed as follows. Assume that every follower has an unconstrained strategy set, that is, $\Omega_i = \mathbb{R}^d$. Then, the implicit gradient is*

$$\frac{d\phi^*\left(\theta\right)}{d\theta} = -\mathbf{P}\left(\phi^*\left(\theta\right), \theta\right)^{-1} \mathbf{Q}\left(\phi^*\left(\theta\right), \theta\right) \tag{26}$$

*where*

$$\mathbf{P}\left(\phi^*\left(\theta\right), \theta\right) = \left[ \frac{\partial}{\partial \phi} \left[ \frac{\partial \mathcal{L}_i\left(\phi_i^*\left(\theta\right); \mathcal{D}_i^{\mathrm{tr}}\right)}{\partial \phi_i} \right]^{\mathrm{T}}_{i \in [N]} + \frac{\partial^2 g\left(\phi^*\left(\theta\right), \theta\right)}{\partial \phi^2} \right]$$

$$\mathbf{Q}\left(\phi^*\left(\theta\right), \theta\right) = \frac{\partial}{\partial \theta} \left[ \frac{\partial g\left(\phi^*\left(\theta\right), \theta\right)}{\partial \phi} \right]^{\mathrm{T}} \tag{27}$$

*Proof.* Let $\hat{\Gamma} = \left\langle f^1, f^2, f^3, \Omega^1, \Omega^2, \Omega^3 \right\rangle$ be a $1 - 1 - 1$ Stackelberg game where $f^1 = F$ is a first leader's utility function and its decision variable $\theta$ in strategy set $\Omega^1 = \mathbb{R}^d$, $f^2 = \sum_{i \in [N]} \left(\frac{\partial f_i}{\partial \phi_i}\right)\left(\phi_i - \hat{\phi}_i\right)$ is a second leader's utility function and its decision variable $\phi$ in strategy set $\Omega^2 = \prod_{i \in \mathbf{F}} \Omega_i\left(\phi_{-i}, \theta\right)$, and $f^3 = -\sum_{i \in [N]} \left(\frac{\partial f_i}{\partial \phi_i}\right)\left(\phi_i - \hat{\phi}_i\right)$ is a follower's utility function and its decision variable $\hat{\phi}$ in strategy set $\Omega^3 = \prod_{i \in \mathbf{F}} \Omega_i\left(\hat{\phi}_{-i}, \theta\right)$. Then, the Stackelberg equilibrium $\left(\theta^*, \phi^*\left(\theta\right), \hat{\phi}^*\left(\theta^*, \phi^*\left(\theta\right)\right)\right)$ of $1 - 1 - 1$ Stackelberg game $\hat{\Gamma}$ that satisfies equations (28) - (30) is also a generalized Stackelberg equilibrium $\left(\theta^*, \phi^*\left(\theta\right)\right)$ of the $1 - N$ generalized Stackelberg game $\Gamma$ (Jo et al., 2023).

$$\theta^* = \arg \min_{\theta \in \mathbb{R}^d} f^1\left(\theta, \phi^*\left(\theta\right)\right) \tag{28}$$

$$\phi^*\left(\theta\right) = \arg \min_{\phi \in \Omega^2(\theta)} f^2\left(\phi, \hat{\phi}^*\left(\phi, \theta\right), \theta\right) \tag{29}$$

$$\hat{\phi}^*\left(\phi, \theta\right) = \arg \min_{\hat{\phi} \in \Omega^3(\theta)} f^3\left(\hat{\phi}, \phi, \theta\right) \tag{30}$$

Now we compute the implicit gradient when every follower of $\Gamma$ has an unconstrained strategy set, i.e., $\Omega_i = \mathbb{R}^d$. The derivative of $f^3$ with respect to $\hat{\phi}$ is a zero vector regardless of $\hat{\phi}$ if the following holds.

$$\frac{d}{d\hat{\phi}} f^3 \left( \hat{\phi}^* \left( \phi, \theta \right), \phi, \theta \right) = \left[ \frac{\partial \mathcal{L}_i \left( \phi_i; \mathcal{D}_i^{\text{tr}} \right)}{\partial \phi_i} + \frac{\partial g \left( \phi, \theta \right)}{\partial \phi_i} \right]_{i \in [N]}^{\text{T}} = \mathbf{0}^{Nd \times 1} \tag{31}$$

Thus, the optimal $\phi^* \left( \theta \right)$ should satisfy equation (31). The derivative of equation (31) with respect to $\theta$ is $\mathbf{0}^{Nd \times d}$.

$$\begin{aligned}
\frac{d}{d\theta} \frac{df^3}{d\hat{\phi}} &= \frac{d}{d\theta} \left[ \frac{\partial \mathcal{L}_i \left( \phi_i^* \left( \theta \right); \mathcal{D}_i^{\text{tr}} \right)}{\partial \phi_i} + \frac{\partial g \left( \phi^* \left( \theta \right), \theta \right)}{\partial \phi_i} \right]_{i \in [N]}^{\text{T}} \\
&= \frac{\partial}{\partial \phi} \left[ \frac{\partial \mathcal{L}_i \left( \phi_i^* \left( \theta \right); \mathcal{D}_i^{\text{tr}} \right)}{\partial \phi_i} \right]_{i \in [N]}^{\text{T}} \frac{d\phi^* \left( \theta \right)}{d\theta} \\
&\quad + \frac{\partial}{\partial \theta} \left[ \frac{\partial g \left( \phi^* \left( \theta \right), \theta \right)}{\partial \phi} \right]^{\text{T}} + \frac{\partial^2 g \left( \phi^* \left( \theta \right), \theta \right)}{\partial \phi^2} \frac{d\phi^* \left( \theta \right)}{d\theta} \\
&= \mathbf{0}^{Nd \times d}
\end{aligned} \tag{32}$$

Therefore, the implicit gradient is as follows.

$$\begin{aligned}
\frac{d\phi^* \left( \theta \right)}{d\theta} &= - \left[ \frac{\partial}{\partial \phi} \left[ \frac{\partial \mathcal{L}_i \left( \phi_i^* \left( \theta \right); \mathcal{D}_i^{\text{tr}} \right)}{\partial \phi_i} \right]_{i \in [N]}^{\text{T}} + \frac{\partial^2 g \left( \phi^* \left( \theta \right), \theta \right)}{\partial \phi^2} \right]^{-1} \\
&\quad \times \frac{\partial}{\partial \theta} \left[ \frac{\partial g \left( \phi^* \left( \theta \right), \theta \right)}{\partial \phi} \right]^{\text{T}}
\end{aligned} \tag{33}$$

The rest of our study focuses on proving the convergence of Algorithm 1 where $\Omega_i = \mathbb{R}^d$ for all task $i$. Even if there are constraints on the strategy set, convergence is guaranteed using the implicit gradient computed by Lemma 3 in a similar manner. To prove the convergence of Algorithm 1 we first define the approximated gradient of the meta loss function $F$ using Lemma 3.

$$\frac{\hat{d}}{d\theta} F \left( \theta, \hat{\phi} \right) = \frac{\partial}{\partial \theta} F \left( \theta, \hat{\phi} \right) + \frac{d\hat{\phi}}{d\theta} \times \frac{\partial}{\partial \phi} F \left( \theta, \hat{\phi} \right) \tag{34}$$

where $\hat{\phi}$ is joint estimated task-specific parameter and

$$\begin{aligned}
\frac{d\hat{\phi}}{d\theta} &= -\mathbf{P} \left( \hat{\phi}, \theta \right)^{-1} \mathbf{Q} \left( \hat{\phi}, \theta \right) \\
\mathbf{P} \left( \hat{\phi}, \theta \right) &= \left[ \frac{\partial}{\partial \phi} \left[ \frac{\partial \mathcal{L}_i \left( \hat{\phi}_i; \mathcal{D}_i^{\text{tr}} \right)}{\partial \phi_i} \right]_{i \in [N]}^{\text{T}} + \frac{\partial^2 g \left( \hat{\phi}, \theta \right)}{\partial \phi^2} \right] \\
\mathbf{Q} \left( \hat{\phi}, \theta \right) &= \frac{\partial}{\partial \theta} \left[ \frac{\partial g \left( \hat{\phi}, \theta \right)}{\partial \phi} \right]^{\text{T}}
\end{aligned} \tag{35}$$

Next, we define a $\delta$-accurate estimation of the optimal joint task-specific parameter $\phi^*$ and an $\epsilon$-accurate estimation of the approximated gradient of the meta loss function $\frac{\hat{d}F}{d\theta}$.

**Definition 1** *Let the joint task-specific parameter $\hat{\phi}$ be a solution estimated by the computing algorithm (e.g. PRGD method). Then, $\hat{\phi}$ is a $\delta$-accurate estimation of the optimal joint task-specific parameter $\phi^*$ if it satisfies the following:*

$$\left\| \hat{\phi} - \phi^* \right\| \leq \delta \tag{36}$$

**Definition 2** *Let $\frac{\hat{d}F}{d\theta}$ be an approximated gradient of the meta loss function. Then, $\hat{h}_\theta$ is an $\epsilon$-accurate estimation of the meta loss function if it satisfies the following:*

$$\left\| \frac{\hat{d}}{d\theta} F\left(\theta, \hat{\phi}\right) - \hat{h}_\theta\left(\theta, \hat{\phi}\right) \right\| \le \epsilon \tag{37}$$

Note that $\hat{\phi}(\theta)$ is a $\delta$-accurate estimation of the variational equilibrium since the PRGD method computes the solution of the variational inequality (Malitsky, 2015). We show that $\hat{\phi}(\theta)$ computed by the PRGD method is also a $\delta$-accurate estimation of the generalized Nash equilibrium by Lemma 2. Now we prove the convergence of the lower-level problem of the $1 - N$ generalized Stackelberg game $\Gamma = \left\langle \{1\}, [N], F, (f_i)_{i\in[N]}, \mathbb{R}^d, (\Omega_i)_{i\in[N]} \right\rangle$ which model the one-step gradient update of meta-parameter for the NashMAML algorithm.

**Theorem 1** *Let $D$ be the diameter of search space of the joint task-specific parameter $\phi = (\phi_i)_{i\in[N]}$ in the inner optimization problem (i.e. $\|\phi - \phi^*(\theta)\| \le D$). Suppose that the projected reflected gradient descent (PRGD) method (Malitsky, 2015) is used to compute the $\delta$-accurate estimation of the optimal joint task-specific parameter $\hat{\phi} = \left(\hat{\phi}_i\right)_{i\in[N]}$ of the generalized Nash equilibrium, which is the convergent point of the inner loop of the NashMAML algorithm. Under Assumption 1, the NashMAML algorithm computes $\hat{\phi}$ with $O\left(\kappa \log\left(D/\delta\right)\right)$ number of iterations, and only $O\left(\text{Mem}\left(\nabla \mathcal{L}_i\right) \kappa \log\left(D/\delta\right)\right)$ memory is required throughout.*

*Proof.* By Theorem 3.3 of (Malitsky, 2015), the PRGD method converges to the $\delta$-accurate estimation of the optimal joint task-specific parameter in $n$ steps, described as follows:

$$\left\| \hat{\phi} - \phi^*(\theta) \right\|^2 \le \gamma^n D^2 \tag{38}$$

where $\gamma = \frac{1 - 2a\mu + \sqrt{1 + 4a^2\mu^2}}{2}$. Under Assumption 1, the number of iteration $n$ to compute $\delta$-accurate estimation of the optimal joint task-specific parameter $\hat{\phi}$ is $-\frac{2}{\log \gamma} \log\left(D/\delta\right)$. Since $a$ is proportional to $1/L$, and $-\frac{2}{\log \gamma}$ is proportional to $\frac{1}{a\mu}$, the number of gradient computations to compute $\hat{\phi}$ is bounded as $\kappa \log\left(D/\delta\right)$. While the process of updating task-specific parameters and computing the implicit gradient of the NashMAML algorithm can be carried out independently for each task-specific parameter, the memory usage is the same as that of the other MAML algorithms (Rajeswaran et al., 2019).

Now we introduce assumptions regarding the utility functions of the leader and followers of the $1 - N$ generalized Stackelberg game $\Gamma = \left\langle \{1\}, [N], F, (f_i)_{i\in[N]}, \mathbb{R}^d, (\Omega_i)_{i\in[N]} \right\rangle$.

**Assumption 2** *We denote the optimal loss function of task $i$ as $\mathcal{L}_i^*(\theta) = \mathcal{L}_i(\theta, \phi^*(\theta))$. Suppose the following holds:*

- *For any $\theta$, $\frac{\partial}{\partial\theta} F$ is Lipschitz continuous with respect to $\phi$ with constant $L_1 > 0$, i.e.*

$$\left\| \frac{\partial}{\partial\theta} F(\theta, \phi_1) - \frac{\partial}{\partial\theta} F(\theta, \phi_2) \right\| \le L_1 \|\phi_1 - \phi_2\|, \forall \phi_1, \phi_2 \tag{39}$$

- *For any $\phi$, $\frac{\partial}{\partial\theta} F$ is Lipschitz continuous with respect to $\theta$ with constant $\bar{L}_1 > 0$, i.e.*

$$\left\| \frac{\partial}{\partial\theta} F(\theta_1, \phi) - \frac{\partial}{\partial\theta} F(\theta_2, \phi) \right\| \le \bar{L}_1 \|\theta_1 - \theta_2\|, \forall \theta_1, \theta_2 \tag{40}$$

- *For any $\theta$, $\frac{\partial}{\partial\phi} F$ is Lipschitz continuous with respect to $\phi$ with constant $L_2 > 0$, i.e.*

$$\left\| \frac{\partial}{\partial\phi} F(\theta, \phi_1) - \frac{\partial}{\partial\phi} F(\theta, \phi_2) \right\| \le L_2 \|\phi_1 - \phi_2\|, \forall \phi_1, \phi_2 \tag{41}$$

- *For any $\phi$, $\frac{\partial}{\partial\phi}F$ is Lipschitz continuous with respect to $\theta$ with constant $\bar{L}_2 > 0$, i.e.*

$$\left\|\frac{\partial}{\partial\phi}F(\theta_1, \phi) - \frac{\partial}{\partial\phi}F(\theta_2, \phi)\right\| \leq \bar{L}_2 \|\theta_1 - \theta_2\|, \forall\theta_1, \theta_2 \tag{42}$$

- *For any $\theta$, and any $\phi$, we have $\left\|\frac{\partial F}{\partial\phi}\right\| \leq C_1$ for some constant $C_1 > 0$.*

- *For any $\theta$, and any $\phi$, we have $\left\|\frac{\partial F}{\partial\theta}\right\| \leq \bar{C}_1$ for some constant $\bar{C}_1 > 0$.*

- *For any $\theta$, $\frac{\partial}{\partial\phi}\left[\frac{\partial\mathcal{L}_i(\hat{\phi}_i; \mathcal{D}_i^{\mathrm{tr}})}{\partial\phi_i}\right]^{\mathrm{T}}_{i\in[N]}$ is Lipschitz continuous with respect to $\phi$ with constant $L_3 > 0$, i.e.*

$$\left\|\frac{\partial}{\partial\phi}\left[\frac{\partial\mathcal{L}_i\left(\phi_i^1; \mathcal{D}_i^{\mathrm{tr}}\right)}{\partial\phi_i}\right]^{\mathrm{T}}_{i\in[N]} - \frac{\partial}{\partial\phi}\left[\frac{\partial\mathcal{L}_i\left(\phi_2^1; \mathcal{D}_i^{\mathrm{tr}}\right)}{\partial\phi_i}\right]^{\mathrm{T}}_{i\in[N]}\right\|$$
$$\leq L_3 \left\|\phi^1 - \phi^2\right\|, \forall\phi^1, \phi^2 \tag{43}$$

- *For any $\theta$, $\frac{\partial^2 g}{\partial\theta\partial\phi}$ is Lipschitz continuous with respect to $\phi$ with constant $L_4 > 0$, i.e.*

$$\left\|\frac{\partial^2}{\partial\theta\partial\phi}g(\theta, \phi_1) - \frac{\partial^2}{\partial\theta\partial\phi}g(\theta, \phi_2)\right\| \leq L_4 \|\phi_1 - \phi_2\|, \forall\phi_1, \phi_2 \tag{44}$$

- *For any $\theta$, $\frac{\partial^2 g}{\partial\phi^2}$ is Lipschitz continuous with respect to $\phi$ with constant $L_5 > 0$, i.e.*

$$\left\|\frac{\partial^2}{\partial\phi^2}g(\theta, \phi_1) - \frac{\partial^2}{\partial\phi^2}g(\theta, \phi_2)\right\| \leq L_5 \|\phi_1 - \phi_2\|, \forall\phi_1, \phi_2 \tag{45}$$

- *The optimal loss function of task $i$, $\mathcal{L}_i^*(\theta)$ is $L_6$-smooth for all task $i$, i.e.*

$$\mathcal{L}_i(\theta_1, \phi^*(\theta_1)) \leq \mathcal{L}_i(\theta_2, \phi^*(\theta_2)) + \left\langle\theta_1 - \theta_2, \frac{d}{d\theta}\mathcal{L}_i(\theta_2, \phi^*(\theta_2))\right\rangle$$
$$+ \frac{L_6}{2}\|\theta_1 - \theta_2\|^2, \forall\theta_1, \theta_2 \tag{46}$$

- *For any $\theta$, $\mathcal{L}_i$ is strongly convex with respect to $\phi_i$ with parameter $\mu_1 > 0$, i.e.*

$$\mu_1\mathbf{I} \preceq \frac{\partial^2\mathcal{L}_i}{\partial\phi_i^2} \tag{47}$$

- *For any $\theta$, $g$ is strongly convex with respect to $\phi$ with parameter $\mu_2 > 0$, i.e.*

$$\mu_2\mathbf{I} \preceq \frac{\partial^2 g}{\partial\phi^2} \tag{48}$$

- *For any $\theta$, and any $\phi$, we have $\left\|\frac{\partial^2 g}{\partial\theta\partial\phi}\right\| \leq C_2$ for some constant $C_2 > 0$.*

The second main result of our paper is that the error between the estimated gradient $\hat{h}_\theta$ computed through back-propagation and $\frac{dF}{d\theta}$ is bounded by a weighted sum of the error in estimating $\phi$ and the error in estimating gradient through back-propagation.

**Lemma 4** *Let $\theta$ be a given meta-parameter, $\phi^*$ be an optimal task-specific parameter, and $\hat{\phi}$ be an estimated task-specific parameter. Under Assumption 2, the following statements hold.*

- *For the same sampling tasks, $\phi^*(\theta)$ is Lipschitz continuous with respect to $\theta$ with constant $\frac{C_2}{\mu_1+\mu_2} > 0$, i.e.*

$$\|\phi^*(\theta_1) - \phi^*(\theta_2)\| \leq \frac{C_2}{\mu_1 + \mu_2}\|\theta_1 - \theta_2\|, \forall\theta_1, \theta_2 \tag{49}$$

- *The difference between the approximated gradient $\frac{\hat{d}F}{d\theta}$ and $\frac{dF}{d\theta}$ is bounded by the error in estimating $\phi^*$. That is,*

$$\left\| \frac{d}{d\theta} F\left(\theta, \phi^*\left(\theta\right)\right) - \frac{\hat{d}}{d\theta} F\left(\theta, \hat{\phi}\left(\theta\right)\right) \right\| \leq C \left\| \phi^*\left(\theta\right) - \hat{\phi}\left(\theta\right) \right\| \tag{50}$$

  *where $C = L_1 + \frac{C_1 L_4 + C_2 L_2}{\mu_1 + \mu_2} + \frac{C_1 C_2 (L_3 + L_5)}{(\mu_1 + \mu_2)^2}$.*

- *The gradient of the optimal $F$ with respect to $\theta$ is Lipschitz continuous in $\theta$ with constant $L_F > 0$, i.e.*

$$\left\| \frac{d}{d\theta} F\left(\theta_1, \phi^*\left(\theta_1\right)\right) - \frac{d}{d\theta} F\left(\theta_2, \phi^*\left(\theta_2\right)\right) \right\| \leq L_F \left\| \theta_1 - \theta_2 \right\| \tag{51}$$

  *where $L_F = \bar{L}_1 + \frac{C_2}{\mu_1 + \mu_2} \left(\bar{L}_2 + C\right)$.*

*Proof.* First, we prove the implicit gradient is bounded. The implicit gradient $\frac{d\phi^*(\theta)}{d\theta}$ is computed as follows by Lemma 3.

$$\frac{d\phi^*\left(\theta\right)}{d\theta} = -\mathbf{P}\left(\phi^*\left(\theta\right), \theta\right)^{-1} \mathbf{Q}\left(\phi^*\left(\theta\right), \theta\right) \tag{52}$$

Under Assumption 2, $\mathbf{P}^{-1}$ and $\mathbf{Q}$ is bounded as follows.

$$\left\| \mathbf{P}\left(\phi^*\left(\theta\right), \theta\right)^{-1} \right\| \leq \frac{1}{\mu_1 + \mu_2}$$
$$\left\| \mathbf{Q}\left(\phi^*\left(\theta\right), \theta\right) \right\| \leq C_2 \tag{53}$$

Thus, the implicit gradient is bounded as $\frac{C_2}{\mu_1 + \mu_2}$.

$$\begin{aligned} \left\| \frac{d\phi^*\left(\theta\right)}{d\theta} \right\| &= \left\| -\mathbf{P}\left(\phi^*\left(\theta\right), \theta\right)^{-1} \mathbf{Q}\left(\phi^*\left(\theta\right), \theta\right) \right\| \\ &\leq \left\| \mathbf{P}\left(\phi^*\left(\theta\right), \theta\right)^{-1} \right\| \times \left\| \mathbf{Q}\left(\phi^*\left(\theta\right), \theta\right) \right\| \\ &\leq \frac{C_2}{\mu_1 + \mu_2} \end{aligned} \tag{54}$$

Now we can prove $\phi^*\left(\theta\right)$ is Lipschitz continuous. The following holds for all $\theta_1, \theta_2$.

$$\begin{aligned} \left\| \phi^*\left(\theta_1\right) - \phi^*\left(\theta_2\right) \right\| &\leq \left\| \frac{d\phi^*\left(\theta\right)}{d\theta} \right\| \left\| \theta_1 - \theta_2 \right\| \\ &\leq \frac{C_2}{\mu_1 + \mu_2} \left\| \theta_1 - \theta_2 \right\| \end{aligned} \tag{55}$$

Next, we prove the difference between the approximated gradient $\frac{\hat{d}F}{d\theta}$ and $\frac{dF}{d\theta}$ is bounded.

$$\begin{aligned} \left\| \frac{d}{d\theta} F\left(\theta, \phi^*\left(\theta\right)\right) - \frac{\hat{d}}{d\theta} F\left(\theta, \hat{\phi}\left(\theta\right)\right) \right\| &= \left\| \mathbf{M}_1 + \mathbf{M}_2 \right\| \\ &= \left\| \mathbf{M}_1 + \mathbf{M}_3 + \mathbf{M}_4 \right\| \\ &= \left\| \mathbf{M}_1 + \mathbf{M}_3 + \left(\mathbf{M}_5 + \mathbf{M}_6\right) \frac{\partial}{\partial \phi} F\left(\theta, \phi^*\left(\theta\right)\right) \right\| \end{aligned} \tag{56}$$

where

$$\mathbf{M}_1 = \frac{\partial}{\partial\theta} F\left(\theta, \phi^*\left(\theta\right)\right) - \frac{\partial}{\partial\theta} F\left(\theta, \hat{\phi}\left(\theta\right)\right)$$

$$\mathbf{M}_2 = \frac{d\phi^*\left(\theta\right)}{d\theta} \frac{\partial}{\partial\phi} F\left(\theta, \phi^*\left(\theta\right)\right) - \frac{d\hat{\phi}\left(\theta\right)}{d\theta} \frac{\partial}{\partial\phi} F\left(\theta, \hat{\phi}\left(\theta\right)\right)$$

$$\mathbf{M}_3 = \frac{d\hat{\phi}\left(\theta\right)}{d\theta} \left( \frac{\partial}{\partial\phi} F\left(\theta, \phi^*\left(\theta\right)\right) - \frac{\partial}{\partial\phi} F\left(\theta, \hat{\phi}\left(\theta\right)\right) \right)$$

$$\mathbf{M}_4 = \left( \frac{d\phi^*\left(\theta\right)}{d\theta} - \frac{d\hat{\phi}\left(\theta\right)}{d\theta} \right) \frac{\partial}{\partial\phi} F\left(\theta, \phi^*\left(\theta\right)\right)$$

$$\mathbf{M}_5 = \left( \mathbf{P}\left(\hat{\phi}\left(\theta\right), \theta\right)^{-1} - \mathbf{P}\left(\phi^*\left(\theta\right), \theta\right)^{-1} \right) \mathbf{Q}\left(\phi^*\left(\theta\right), \theta\right)$$

$$\mathbf{M}_6 = \mathbf{P}\left(\hat{\phi}\left(\theta\right), \theta\right)^{-1} \left( \mathbf{Q}\left(\hat{\phi}\left(\theta\right), \theta\right) - \mathbf{Q}\left(\phi^*\left(\theta\right), \theta\right) \right) \tag{57}$$

Under Assumption 2, each term of equation (56) satisfies the following inequalities. Because $\frac{\partial}{\partial\theta}F$ is Lipschitz continuous,

$$\|\mathbf{M}_1\| \le L_1 \left\| \phi^*\left(\theta\right) - \hat{\phi}\left(\theta\right) \right\| \tag{58}$$

Because $\frac{\partial}{\partial\phi}F$ is Lipschitz continuous,

$$
\begin{aligned}
\|\mathbf{M}_3\| &\le \left\| \frac{d\hat{\phi}\left(\theta\right)}{d\theta} \right\| \left\| \frac{\partial}{\partial\phi} F\left(\theta, \phi^*\left(\theta\right)\right) - \frac{\partial}{\partial\phi} F\left(\theta, \hat{\phi}\left(\theta\right)\right) \right\| \\
&\le \left\| -\mathbf{P}\left(\hat{\phi}\left(\theta\right), \theta\right)^{-1} \mathbf{Q}\left(\hat{\phi}\left(\theta\right), \theta\right) \right\| \times L_2 \left\| \phi^*\left(\theta\right) - \hat{\phi}\left(\theta\right) \right\| \\
&\le \frac{C_2 L_2}{\mu_1 + \mu_2} \left\| \phi^*\left(\theta\right) - \hat{\phi}\left(\theta\right) \right\|
\end{aligned} \tag{59}
$$

Because $\mathbf{P}$ is Lipschitz continuous,

$$
\begin{aligned}
\|\mathbf{M}_5\| &\le \left\| \mathbf{P}\left(\hat{\phi}\left(\theta\right), \theta\right)^{-1} - \mathbf{P}\left(\phi^*\left(\theta\right), \theta\right)^{-1} \right\| \left\| \frac{\partial^2 g}{\partial\theta\partial\phi} \right\| \\
&= \left\| \mathbf{P}\left(\phi^*\left(\theta\right), \theta\right)^{-1} \left( \mathbf{P}\left(\phi^*\left(\theta\right), \theta\right) - \mathbf{P}\left(\hat{\phi}\left(\theta\right), \theta\right) \right) \mathbf{P}\left(\hat{\phi}\left(\theta\right), \theta\right)^{-1} \right\| \left\| \frac{\partial^2 g}{\partial\theta\partial\phi} \right\| \\
&\le \left\| \mathbf{P}\left(\phi^*\left(\theta\right), \theta\right)^{-1} \right\| \left\| \mathbf{P}\left(\phi^*\left(\theta\right), \theta\right) - \mathbf{P}\left(\hat{\phi}\left(\theta\right), \theta\right) \right\| \left\| \mathbf{P}\left(\hat{\phi}\left(\theta\right), \theta\right)^{-1} \right\| \left\| \frac{\partial^2 g}{\partial\theta\partial\phi} \right\| \\
&\le \frac{C_2}{\left(\mu_1 + \mu_2\right)^2} \left\| \frac{\partial}{\partial\phi} \left[ \frac{\partial\mathcal{L}_i\left(\phi_i^*\left(\theta\right); \mathcal{D}_i^{\mathrm{tr}}\right)}{\partial\phi_i} \right]_{i\in[N]}^{\mathrm{T}} - \frac{\partial}{\partial\phi} \left[ \frac{\partial\mathcal{L}_i\left(\hat{\phi}_i\left(\theta\right); \mathcal{D}_i^{\mathrm{tr}}\right)}{\partial\phi_i} \right]_{i\in[N]}^{\mathrm{T}} \right\| \\
&\quad + \frac{C_2}{\left(\mu_1 + \mu_2\right)^2} \left\| \frac{\partial^2 g\left(\phi^*\left(\theta\right), \theta\right)}{\partial\phi^2} - \frac{\partial^2 g\left(\hat{\phi}\left(\theta\right), \theta\right)}{\partial\phi^2} \right\| \\
&\le \frac{C_2\left(L_3 + L_5\right)}{\left(\mu_1 + \mu_2\right)^2} \left\| \phi^*\left(\theta\right) - \hat{\phi}\left(\theta\right) \right\|
\end{aligned} \tag{60}
$$

Because $\frac{\partial^2 g}{\partial \theta \partial \phi}$ is Lipschitz continuous,

$$\left\| (\mathbf{M}_5 + \mathbf{M}_6) \frac{\partial}{\partial \phi} F(\theta, \phi^*(\theta)) \right\| \leq (\|\mathbf{M}_5\| + \|\mathbf{M}_6\|) \left\| \frac{\partial}{\partial \phi} F(\theta, \phi^*(\theta)) \right\|$$
$$\leq \frac{C_1 C_2 (L_3 + L_5)}{(\mu_1 + \mu_2)^2} \left\| \phi^*(\theta) - \hat{\phi}(\theta) \right\|$$
$$+ \frac{L_4 C_1}{\mu_1 + \mu_2} \left\| \phi^*(\theta) - \hat{\phi}(\theta) \right\| \tag{61}$$

Thus, the equation (56) is expanded as follows.

$$\left\| \frac{d}{d\theta} F(\theta, \phi^*(\theta)) - \frac{\hat{d}}{d\theta} F(\theta, \hat{\phi}(\theta)) \right\| \leq \left\| \mathbf{M}_1 + \mathbf{M}_3 + (\mathbf{M}_5 + \mathbf{M}_6) \frac{\partial}{\partial \phi} F(\theta, \phi^*(\theta)) \right\|$$
$$\leq \|\mathbf{M}_1\| + \|\mathbf{M}_3\| + \left\| (\mathbf{M}_5 + \mathbf{M}_6) \frac{\partial}{\partial \phi} F(\theta, \phi^*(\theta)) \right\|$$
$$\leq C \left\| \phi^*(\theta) - \hat{\phi}(\theta) \right\| \tag{62}$$

where

$$C = L_1 + \frac{C_1 L_4 + C_2 L_2}{\mu_1 + \mu_2} + \frac{C_1 C_2 (L_3 + L_5)}{(\mu_1 + \mu_2)^2} \tag{63}$$

Finally, we prove the gradient of the optimal $F$ with respect to $\theta$ is Lipschitz continuous in $\theta$.

$$\left\| \frac{d}{d\theta} F(\theta_1, \phi^*(\theta_1)) - \frac{d}{d\theta} F(\theta_2, \phi^*(\theta_2)) \right\| \leq \left\| \frac{d}{d\theta} F(\theta_1, \phi^*(\theta_1)) - \frac{\hat{d}}{d\theta} F(\theta_1, \phi^*(\theta_2)) \right\|$$
$$+ \left\| \frac{\hat{d}}{d\theta} F(\theta_1, \phi^*(\theta_2)) - \frac{d}{d\theta} F(\theta_2, \phi^*(\theta_2)) \right\| \tag{64}$$

The first term of equation (64) is expanded as follows using equations (49), (50).

$$\left\| \frac{d}{d\theta} F(\theta_1, \phi^*(\theta_1)) - \frac{\hat{d}}{d\theta} F(\theta_1, \phi^*(\theta_2)) \right\| \leq C \|\phi^*(\theta_1) - \phi^*(\theta_2)\|$$
$$\leq \frac{C C_2}{\mu_1 + \mu_2} \|\theta_1 - \theta_2\| \tag{65}$$

Under Assumption 2, the second term of equation (64) is expanded as

$$\left\| \frac{\hat{d}}{d\theta} F(\theta_1, \phi^*(\theta_2)) - \frac{d}{d\theta} F(\theta_2, \phi^*(\theta_2)) \right\| \leq \left\| \frac{\partial}{\partial \theta} F(\theta_1, \phi^*(\theta_2)) - \frac{\partial}{\partial \theta} F(\theta_2, \phi^*(\theta_2)) \right\|$$
$$+ \left\| \frac{d\phi^*(\theta_2)}{d\theta} \right\| \left\| \frac{\partial}{\partial \phi} F(\theta_1, \phi^*(\theta_2)) - \frac{\partial}{\partial \phi} F(\theta_2, \phi^*(\theta_2)) \right\|$$
$$\leq \bar{L}_1 \|\theta_1 - \theta_2\| + \frac{C_2}{\mu_1 + \mu_2} \bar{L}_2 \|\theta_1 - \theta_2\| \tag{66}$$

Now we prove the optimal $F^*$ is Lipschitz continuous with respect to $\theta$ with constant $L_F$ by equations (65) and (66).

$$\left\| \frac{d}{d\theta} F(\theta_1, \phi^*(\theta_1)) - \frac{d}{d\theta} F(\theta_2, \phi^*(\theta_2)) \right\| \leq L_F \|\theta_1 - \theta_2\| \tag{67}$$

where $L_F = \bar{L}_1 + \frac{C_2}{\mu_1 + \mu_2} (\bar{L}_2 + C)$.

**Theorem 2** *Let $\theta$ be a given meta-parameter, $\phi^*$ be an optimal task-specific parameter, $\hat{\phi}$ be a $\delta$-accurate estimated task-specific parameter, and $\hat{h}_\theta$ be an $\epsilon$-accurate estimated gradient of $F$ with*

respect to $\theta$ computed through back-propagation. Under Assumption 2, the difference between the $\epsilon$-accurate estimated gradient $\hat{h}_\theta$ and the gradient of the optimal meta loss function $F$ with respect to $\theta$, $\frac{dF}{d\theta}$, is bounded by the weighted sum of the error in estimating $\phi^*$ and the error in estimating the gradient through back-propagation. That is,

$$\left\| \frac{d}{d\theta} F\left(\theta, \phi^*\left(\theta\right)\right) - \hat{h}_\theta\left(\theta, \hat{\phi}\right) \right\| \leq C \left\| \phi^*\left(\theta\right) - \hat{\phi} \right\| + \left\| \frac{\hat{d}}{d\theta} F\left(\theta, \hat{\phi}\right) - \hat{h}_\theta\left(\theta, \hat{\phi}\right) \right\|$$

$$\leq C\delta + \epsilon \tag{68}$$

where $C = L_1 + \frac{C_1 L_4 + C_2 L_2}{\mu_1 + \mu_2} + \frac{C_1 C_2 (L_3 + L_5)}{(\mu_1 + \mu_2)^2}$.

*Proof.* Because the triangle inequality $\|a + b\| \leq \|a\| + \|b\|$ holds,

$$\left\| \frac{d}{d\theta} F\left(\theta, \phi^*\left(\theta\right)\right) - \hat{h}_\theta\left(\theta, \hat{\phi}\right) \right\| = \left\| \frac{d}{d\theta} F\left(\theta, \phi^*\left(\theta\right)\right) - \frac{\hat{d}}{d\theta} F\left(\theta, \hat{\phi}\right) + \frac{\hat{d}}{d\theta} F\left(\theta, \hat{\phi}\right) - \hat{h}_\theta\left(\theta, \hat{\phi}\right) \right\|$$

$$\leq \left\| \frac{d}{d\theta} F\left(\theta, \phi^*\left(\theta\right)\right) - \frac{\hat{d}}{d\theta} F\left(\theta, \hat{\phi}\right) \right\| + \left\| \frac{\hat{d}}{d\theta} F\left(\theta, \hat{\phi}\right) - \hat{h}_\theta\left(\theta, \hat{\phi}\right) \right\|$$

$$\leq \left\| \frac{d}{d\theta} F\left(\theta, \phi^*\left(\theta\right)\right) - \frac{\hat{d}}{d\theta} F\left(\theta, \hat{\phi}\right) \right\| + \epsilon \tag{69}$$

By Lemma 4, equation (69) is expanded as follows.

$$\left\| \frac{d}{d\theta} F\left(\theta, \phi^*\left(\theta\right)\right) - \hat{h}_\theta\left(\theta, \hat{\phi}\right) \right\| \leq \left\| \frac{d}{d\theta} F\left(\theta, \phi^*\left(\theta\right)\right) - \frac{\hat{d}}{d\theta} F\left(\theta, \hat{\phi}\right) \right\| + \epsilon$$

$$\leq C \left\| \phi^*\left(\theta\right) - \hat{\phi} \right\| + \epsilon$$

$$\leq C\delta + \epsilon \tag{70}$$

where $C = L_1 + \frac{C_1 L_4 + C_2 L_2}{\mu_1 + \mu_2} + \frac{C_1 C_2 (L_3 + L_5)}{(\mu_1 + \mu_2)^2}$.

For the rest of our paper, we discuss the expected utility function with respect to task sampling in order to prove the convergence of the NashMAML algorithm, and its convergent point is an optimal solution of the stochastic optimization problem described in equations (8) and (9). First, we describe the assumptions and lemmas required to prove the convergence. We denote $\mathbb{E}_{\mathcal{T}_i \sim p(\mathcal{T})}\left[\cdot\right]$ as $\mathbb{E}\left[\cdot\right]$ in the remaining part.

**Assumption 3** *Let $\theta^k$ be a meta-parameter, $\phi^*\left(\theta^k\right)$ be an optimal joint task-specific parameter, and $\hat{\phi}\left(\theta^k\right)$ be an estimated joint task-specific parameter of the $k$-th updated meta-parameter. For any $k \geq 0$, there exists a non-increasing sequence $\{b_k\}_{k \geq 0}$, $\{\sigma_k\}_{k \geq 0}$, and $\{\bar{\sigma}_k\}_{k \geq 0}$ which converge to 0 that satisfies the following.*

- *Let $\hat{\phi}$ be an estimated joint task-specific parameter. Then, the expectation of an estimated gradient $\hat{h}_\theta\left(\theta, \hat{\phi}\left(\theta\right)\right)$ is as follows.*

$$\mathbb{E}\left[ \hat{h}_\theta\left(\theta^k, \hat{\phi}\left(\theta^k\right)\right) \right] = \mathbb{E}\left[ \frac{\hat{d}}{d\theta} F\left(\theta^k, \hat{\phi}\left(\theta^k\right)\right) \right] + B_k, \|B_k\| \leq b_k \tag{71}$$

- *The norm-variance of an estimated gradient $\hat{h}_\theta\left(\theta, \hat{\phi}\left(\theta\right)\right)$ is bounded, i.e.*

$$\mathbb{E}\left[ \left\| \hat{h}_\theta\left(\theta^k, \hat{\phi}\left(\theta^k\right)\right) - \mathbb{E}\left[ \hat{h}_\theta\left(\theta^k, \hat{\phi}\left(\theta^k\right)\right) \right] \right\|^2 \right] \leq \sigma_k^2 \tag{72}$$

- *The norm-variance of an optimal gradient $\frac{d}{d\theta} F\left(\theta^k, \phi^*\left(\theta^k\right)\right)$ is bounded, i.e.*

$$\mathbb{E}\left[ \left\| \frac{d}{d\theta} F\left(\theta^k, \phi^*\left(\theta^k\right)\right) - \mathbb{E}\left[ \frac{d}{d\theta} F\left(\theta^k, \phi^*\left(\theta^k\right)\right) \right] \right\|^2 \right] \leq \bar{\sigma}_k^2 \tag{73}$$

**Lemma 5** *The expectation of the square norm of an estimated gradient $\hat{h}_\theta \left( \theta, \hat{\phi} \left( \theta \right) \right)$ is bounded, i.e.*

$$
\mathbb{E} \left[ \left\| \hat{h}_\theta \left( \theta^k, \hat{\phi} \left( \theta^k \right) \right) \right\|^2 \right] \leq 4C^2 \mathbb{E} \left[ \left\| \hat{\phi} \left( \theta^k \right) - \phi^* \left( \theta^k \right) \right\| \right]^2
$$
$$
+ \sigma_k^2 + 2b_k^2 + 4 \left( \bar{C}_1 + \frac{C_1 C_2}{\mu_1 + \mu_2} \right)^2 \tag{74}
$$

*where $C = L_1 + \frac{C_1 L_4 + C_2 L_2}{\mu_1 + \mu_2} + \frac{C_1 C_2 (L_3 + L_5)}{(\mu_1 + \mu_2)^2}$.*

*Proof.* We denote $\hat{h}_\theta \left( \theta, \hat{\phi} \left( \theta \right) \right)$ as $\hat{h}_\theta \left( \theta \right)$. Then, the expectation of the square norm of the estimated gradient $\hat{h}_\theta \left( \theta \right)$ is as follows.

$$
\mathbb{E} \left[ \left\| \hat{h}_\theta \left( \theta \right) \right\|^2 \right] = \mathbb{E} \left[ \left\| \hat{h}_\theta \left( \theta \right) \right\|^2 \right] + 2 \left\| \mathbb{E} \left[ \hat{h}_\theta \left( \theta \right) \right] \right\|^2 - 2 \left\langle \mathbb{E} \left[ \hat{h}_\theta \left( \theta \right) \right], \mathbb{E} \left[ \hat{h}_\theta \left( \theta \right) \right] \right\rangle
$$
$$
= \mathbb{E} \left[ \left\| \hat{h}_\theta \left( \theta \right) \right\|^2 \right] + 2 \left\| \mathbb{E} \left[ \hat{h}_\theta \left( \theta \right) \right] \right\|^2 - 2 \mathbb{E} \left\langle \hat{h}_\theta \left( \theta \right), \mathbb{E} \left[ \hat{h}_\theta \left( \theta \right) \right] \right\rangle
$$
$$
= \mathbb{E} \left[ \left\| \hat{h}_\theta \left( \theta \right) - \mathbb{E} \left[ \hat{h}_\theta \left( \theta \right) \right] \right\|^2 \right] + \left\| \mathbb{E} \left[ \hat{h}_\theta \left( \theta \right) \right] \right\|^2 \tag{75}
$$

Substituting $\theta$ with $\theta^k$ of equation (75). Because $\|a + b\|^2 \leq 2 \|a\|^2 + 2 \|b\|^2$, equation (75) is expanded as follows under Assumption 3.

$$
\mathbb{E} \left[ \left\| \hat{h}_\theta \left( \theta^k \right) \right\|^2 \right] = \mathbb{E} \left[ \left\| \hat{h}_\theta \left( \theta^k \right) - \mathbb{E} \left[ \hat{h}_\theta \left( \theta^k \right) \right] \right\|^2 \right] + \left\| \mathbb{E} \left[ \hat{h}_\theta \left( \theta^k \right) \right] \right\|^2
$$
$$
\leq \sigma_k^2 + \left\| \mathbb{E} \left[ \frac{\hat{d}}{d\theta} F \left( \theta^k, \hat{\phi} \left( \theta^k \right) \right) \right] + B_k \right\|^2
$$
$$
\leq \sigma_k^2 + 2b_k^2 + 2 \left\| \mathbb{E} \left[ \frac{\hat{d}}{d\theta} F \left( \theta^k, \hat{\phi} \left( \theta^k \right) \right) \right] \right\|^2
$$
$$
\leq \sigma_k^2 + 2b_k^2 + 4 \left\| \mathbb{E} \left[ \frac{d}{d\theta} F \left( \theta^k, \phi^* \left( \theta^k \right) \right) \right] \right\|^2
$$
$$
+ 4 \left\| \mathbb{E} \left[ \frac{\hat{d}}{d\theta} F \left( \theta^k, \hat{\phi} \left( \theta^k \right) \right) \right] - \mathbb{E} \left[ \frac{d}{d\theta} F \left( \theta^k, \phi^* \left( \theta^k \right) \right) \right] \right\|^2 \tag{76}
$$

Because the norm is convex, $\left\| \mathbb{E} \left[ \cdot \right] \right\|^2 \leq \mathbb{E} \left[ \left\| \cdot \right\| \right]^2$ by Jensen's inequality.

$$
\mathbb{E} \left[ \left\| \hat{h}_\theta \left( \theta^k \right) \right\|^2 \right] \leq \sigma_k^2 + 2b_k^2 + 4 \left\| \mathbb{E} \left[ \frac{d}{d\theta} F \left( \theta^k, \phi^* \left( \theta^k \right) \right) \right] \right\|^2
$$
$$
+ 4 \left\| \mathbb{E} \left[ \frac{\hat{d}}{d\theta} F \left( \theta^k, \hat{\phi} \left( \theta^k \right) \right) - \frac{d}{d\theta} F \left( \theta^k, \phi^* \left( \theta^k \right) \right) \right] \right\|^2
$$
$$
\leq \sigma_k^2 + 2b_k^2 + 4 \mathbb{E} \left[ \left\| \frac{d}{d\theta} F \left( \theta^k, \phi^* \left( \theta^k \right) \right) \right\| \right]^2
$$
$$
+ 4 \mathbb{E} \left[ \left\| \frac{\hat{d}}{d\theta} F \left( \theta^k, \hat{\phi} \left( \theta^k \right) \right) - \frac{d}{d\theta} F \left( \theta^k, \phi^* \left( \theta^k \right) \right) \right\| \right]^2 \tag{77}
$$

Because $\frac{dF}{d\theta}\left(\theta,\phi^*\left(\theta\right)\right)=\frac{\partial}{\partial\theta}F\left(\theta,\phi^*\left(\theta\right)\right)+\frac{d\phi^*\left(\theta\right)}{d\theta}\times\frac{\partial}{\partial\phi}F\left(\theta,\phi^*\left(\theta\right)\right)$, the third term and fourth term of equation (77) are expanded as follows by Lemma 4 and Assumption 2.

$$\mathbb{E}\left[\left\|\frac{d}{d\theta}F\left(\theta^k,\phi^*\left(\theta^k\right)\right)\right\|\right]\leq\mathbb{E}\left[\left\|\frac{\partial}{\partial\theta}F\left(\theta^k,\phi^*\left(\theta^k\right)\right)\right\|\right]$$
$$+\mathbb{E}\left[\left\|\frac{d\phi^*\left(\theta^k\right)}{d\theta}\right\|\left\|\frac{\partial}{\partial\phi}F\left(\theta^k,\phi^*\left(\theta^k\right)\right)\right\|\right]$$
$$\leq\bar{C}_1+\frac{C_1C_2}{\mu_1+\mu_2}\tag{78}$$

$$\mathbb{E}\left[\left\|\frac{\hat{d}}{d\theta}F\left(\theta^k,\hat{\phi}\left(\theta^k\right)\right)-\frac{d}{d\theta}F\left(\theta^k,\phi^*\left(\theta^k\right)\right)\right\|\right]\leq C\mathbb{E}\left[\left\|\hat{\phi}\left(\theta^k\right)-\phi^*\left(\theta^k\right)\right\|\right]\tag{79}$$

where $C=L_1+\frac{C_1L_4+C_2L_2}{\mu_1+\mu_2}+\frac{C_1C_2(L_3+L_5)}{(\mu_1+\mu_2)^2}$. Combining equations (77), (78) and (79), we derive the bound of the expectation of the estimated gradient.

$$\mathbb{E}\left[\left\|\hat{h}_\theta\left(\theta^k\right)\right\|^2\right]\leq\sigma_k^2+2b_k^2+4\mathbb{E}\left[\left\|\frac{d}{d\theta}F\left(\theta^k,\phi^*\left(\theta^k\right)\right)\right\|\right]^2$$
$$+4\mathbb{E}\left[\left\|\frac{\hat{d}}{d\theta}F\left(\theta^k,\hat{\phi}\left(\theta^k\right)\right)-\frac{d}{d\theta}F\left(\theta^k,\phi^*\left(\theta^k\right)\right)\right\|\right]^2$$
$$\leq\sigma_k^2+2b_k^2+4\left(\bar{C}_1+\frac{C_1C_2}{\mu_1+\mu_2}\right)^2+4C^2\mathbb{E}\left[\left\|\hat{\phi}\left(\theta^k\right)-\phi^*\left(\theta^k\right)\right\|\right]^2\tag{80}$$

**Lemma 6** *Let $\mathcal{L}\left(\theta\right)$ be an expected optimal meta loss function, that is, $\mathcal{L}\left(\theta\right)=\mathbb{E}\left[F\left(\theta,\phi^*\left(\theta\right)\right)\right]$. Then, $\mathcal{L}$ satisfies the following equation.*

$$\mathcal{L}\left(\theta^{k+1}\right)-\mathcal{L}\left(\theta^k\right)\leq\left(\frac{L_6}{2}-\frac{1}{2\beta}\right)\mathbb{E}\left[\left\|\theta^{k+1}-\theta^k\right\|^2\right]$$
$$+4\beta C^2\mathbb{E}\left[\left\|\phi^*\left(\theta^k\right)-\hat{\phi}\left(\theta^k\right)\right\|\right]^2+4\beta\bar{\sigma}_k^2+2\beta b_k^2+\beta\sigma_k^2\tag{81}$$

*where $C=L_1+\frac{C_1L_4+C_2L_2}{\mu_1+\mu_2}+\frac{C_1C_2(L_3+L_5)}{(\mu_1+\mu_2)^2}$.*

*Proof.* Under Assumption 2, the task $i$'s optimal loss function $\mathcal{L}_i^*\left(\theta\right)=\mathcal{L}_i\left(\theta,\phi^*\left(\theta\right)\right)$ is $L_6$-smooth.

$$\mathcal{L}_i^*\left(\theta^{k+1}\right)\leq\mathcal{L}_i^*\left(\theta^k\right)+\left\langle\theta^{k+1}-\theta^k,\frac{d}{d\theta}\mathcal{L}_i^*\left(\theta^k\right)\right\rangle+\frac{L_6}{2}\left\|\theta^{k+1}-\theta^k\right\|^2\tag{82}$$

Summing up equation (82) for all tasks $i\in[N]$ and divide it by $N$, we obtain

$$F^*\left(\theta^{k+1}\right)\leq F^*\left(\theta^k\right)+\left\langle\theta^{k+1}-\theta^k,\frac{d}{d\theta}F^*\left(\theta^k\right)\right\rangle+\frac{L_6}{2}\left\|\theta^{k+1}-\theta^k\right\|^2\tag{83}$$

where $F^*\left(\theta\right)=F\left(\theta,\phi^*\left(\theta\right)\right)$ is the optimal meta loss function. Then, the difference of the optimal meta loss function $F^*\left(\theta^{k+1}\right)-F^*\left(\theta^k\right)$ is as follows.

$$F^*\left(\theta^{k+1}\right)-F^*\left(\theta^k\right)\leq\left\langle\theta^{k+1}-\theta^k,\frac{d}{d\theta}F^*\left(\theta^k\right)\right\rangle+\frac{L_6}{2}\left\|\theta^{k+1}-\theta^k\right\|^2$$
$$=\left\langle\theta^{k+1}-\theta^k,\frac{d}{d\theta}F^*\left(\theta^k\right)-\mathbb{E}\left[\frac{\hat{d}}{d\theta}F\left(\theta^k,\hat{\phi}\left(\theta^k\right)\right)\right]-B_k\right\rangle$$
$$+\left\langle\theta^{k+1}-\theta^k,\mathbb{E}\left[\frac{\hat{d}}{d\theta}F\left(\theta^k,\hat{\phi}\left(\theta^k\right)\right)\right]+B_k\right\rangle+\frac{L_6}{2}\left\|\theta^{k+1}-\theta^k\right\|^2\tag{84}$$

Because the meta-parameter of the NashMAML algorithm is updated as $\theta^{k+1} = \theta^k - \beta \hat{h}_\theta \left( \theta^k, \hat{\phi} \left( \theta^k \right) \right)$, that is the difference is $\theta^{k+1} - \theta^k = -\beta \hat{h}_\theta \left( \theta^k, \hat{\phi} \left( \theta^k \right) \right)$. Multiplying both sides by $\theta^{k+1} - \theta^k$ and simplifying, we get

$$\left\langle \theta^{k+1} - \theta^k, \hat{h}_\theta \left( \theta^k, \hat{\phi} \left( \theta^k \right) \right) \right\rangle = -\frac{1}{\beta} \left\| \theta^{k+1} - \theta^k \right\|^2 \tag{85}$$

By combining equations (84) and (85),

$$\begin{aligned}
F^* \left( \theta^{k+1} \right) - F^* \left( \theta^k \right) &\leq \left\langle \theta^{k+1} - \theta^k, \frac{d}{d\theta} F^* \left( \theta^k \right) - \mathbb{E} \left[ \frac{\hat{d}}{d\theta} F \left( \theta^k, \hat{\phi} \left( \theta^k \right) \right) \right] - B_k \right\rangle \\
&\quad + \left\langle \theta^{k+1} - \theta^k, \mathbb{E} \left[ \frac{\hat{d}}{d\theta} F \left( \theta^k, \hat{\phi} \left( \theta^k \right) \right) \right] + B_k - \hat{h}_\theta \left( \theta^k, \hat{\phi} \left( \theta^k \right) \right) \right\rangle \\
&\quad - \frac{1}{\beta} \left\| \theta^{k+1} - \theta^k \right\|^2 + \frac{L_6}{2} \left\| \theta^{k+1} - \theta^k \right\|^2
\end{aligned} \tag{86}$$

Because $\langle a, b \rangle \leq \frac{1}{2c} \|a\|^2 + \frac{c}{2} \|b\|^2$ for some constant $c$, the following equation holds under the definition of $\hat{h}_\theta \left( \theta^k, \hat{\phi} \left( \theta^k \right) \right)$.

$$\begin{aligned}
F^* \left( \theta^{k+1} \right) - F^* \left( \theta^k \right) &\leq \frac{1}{2c_1} \left\| \frac{d}{d\theta} F^* \left( \theta^k \right) - \mathbb{E} \left[ \frac{\hat{d}}{d\theta} F \left( \theta^k, \hat{\phi} \left( \theta^k \right) \right) \right] - B_k \right\|^2 \\
&\quad + \frac{1}{2c_2} \left\| \mathbb{E} \left[ \hat{h}_\theta \left( \theta^k, \hat{\phi} \left( \theta^k \right) \right) \right] - \hat{h}_\theta \left( \theta^k, \hat{\phi} \left( \theta^k \right) \right) \right\|^2 \\
&\quad + \left( \frac{c_1 + c_2 + L_6}{2} - \frac{1}{\beta} \right) \left\| \theta^{k+1} - \theta^k \right\|^2
\end{aligned} \tag{87}$$

Under Assumption 3, the expectation of equation (87) is as follows.

$$\begin{aligned}
\mathcal{L} \left( \theta^{k+1} \right) - \mathcal{L} \left( \theta^k \right) &= \mathbb{E} \left[ F \left( \theta^{k+1}, \phi^* \left( \theta^{k+1} \right) \right) \right] - \mathbb{E} \left[ F \left( \theta^k, \phi^* \left( \theta^k \right) \right) \right] \\
&= \mathbb{E} \left[ F^* \left( \theta^{k+1} \right) - F^* \left( \theta^k \right) \right] \\
&\leq \frac{1}{2c_1} \mathbb{E} \left[ \left\| \frac{d}{d\theta} F^* \left( \theta^k \right) - \mathbb{E} \left[ \frac{\hat{d}}{d\theta} F \left( \theta^k, \hat{\phi} \left( \theta^k \right) \right) \right] - B_k \right\|^2 \right] \\
&\quad + \frac{\sigma_k^2}{2c_2} + \left( \frac{c_1 + c_2 + L_6}{2} - \frac{1}{\beta} \right) \mathbb{E} \left[ \left\| \theta^{k+1} - \theta^k \right\|^2 \right] \\
&\leq \frac{2}{c_1} \mathbb{E} \left[ \left\| \frac{d}{d\theta} F^* \left( \theta^k \right) - \mathbb{E} \left[ \frac{d}{d\theta} F^* \left( \theta^k \right) \right] \right\|^2 \right] \\
&\quad + \frac{2}{c_1} \mathbb{E} \left[ \left\| \mathbb{E} \left[ \frac{d}{d\theta} F^* \left( \theta^k \right) \right] - \mathbb{E} \left[ \frac{\hat{d}}{d\theta} F \left( \theta^k, \hat{\phi} \left( \theta^k \right) \right) \right] \right\|^2 \right] \\
&\quad + \frac{b_k^2}{c_1} + \frac{\sigma_k^2}{2c_2} + \left( \frac{c_1 + c_2 + L_6}{2} - \frac{1}{\beta} \right) \mathbb{E} \left[ \left\| \theta^{k+1} - \theta^k \right\|^2 \right]
\end{aligned} \tag{88}$$

Let $\frac{\hat{d}}{d\theta}F\left(\theta^k\right) = \frac{\hat{d}}{d\theta}F\left(\theta^k, \hat{\phi}\left(\theta^k\right)\right)$. By Lemma 4 and Jensen's inequality, the first term and second term of equation (88) is as follows under Assumption 3.

$$\mathbb{E}\left[\left\|\frac{d}{d\theta}F^*\left(\theta^k\right) - \mathbb{E}\left[\frac{d}{d\theta}F^*\left(\theta^k\right)\right]\right\|^2\right] \leq \bar{\sigma}_k^2 \tag{89}$$

$$\mathbb{E}\left[\left\|\mathbb{E}\left[\frac{d}{d\theta}F^*\left(\theta^k\right)\right] - \mathbb{E}\left[\frac{\hat{d}}{d\theta}F\left(\theta^k\right)\right]\right\|^2\right] = \mathbb{E}\left[\left\|\mathbb{E}\left[\frac{d}{d\theta}F^*\left(\theta^k\right) - \frac{\hat{d}}{d\theta}F\left(\theta^k\right)\right]\right\|^2\right]$$

$$\leq \mathbb{E}\left[\mathbb{E}\left[\left\|\frac{d}{d\theta}F^*\left(\theta^k\right) - \frac{\hat{d}}{d\theta}F\left(\theta^k\right)\right\|^2\right]\right]$$

$$= \mathbb{E}\left[\left\|\frac{d}{d\theta}F^*\left(\theta^k\right) - \frac{\hat{d}}{d\theta}F\left(\theta^k\right)\right\|^2\right]$$

$$\leq C^2\mathbb{E}\left[\left\|\phi^*\left(\theta^k\right) - \hat{\phi}\left(\theta^k\right)\right\|\right]^2 \tag{90}$$

Combining equations (88), (89), and (90), the difference of optimal meta loss function is expanded as follows.

$$\mathcal{L}\left(\theta^{k+1}\right) - \mathcal{L}\left(\theta^k\right) \leq \frac{2}{c_1}\mathbb{E}\left[\left\|\frac{d}{d\theta}F^*\left(\theta^k\right) - \mathbb{E}\left[\frac{d}{d\theta}F^*\left(\theta^k\right)\right]\right\|^2\right]$$

$$+ \frac{2}{c_1}\mathbb{E}\left[\left\|\mathbb{E}\left[\frac{d}{d\theta}F^*\left(\theta^k\right)\right] - \mathbb{E}\left[\frac{\hat{d}}{d\theta}F\left(\theta^k, \hat{\phi}\left(\theta^k\right)\right)\right]\right\|^2\right]$$

$$+ \frac{b_k^2}{c_1} + \frac{\sigma_k^2}{2c_2} + \left(\frac{c_1 + c_2 + L_6}{2} - \frac{1}{\beta}\right)\mathbb{E}\left[\left\|\theta^{k+1} - \theta^k\right\|^2\right]$$

$$\leq \left(\frac{c_1 + c_2 + L_6}{2} - \frac{1}{\beta}\right)\mathbb{E}\left[\left\|\theta^{k+1} - \theta^k\right\|^2\right]$$

$$+ \frac{2C^2}{c_1}\mathbb{E}\left[\left\|\phi^*\left(\theta^k\right) - \hat{\phi}\left(\theta^k\right)\right\|\right]^2 + \frac{2\bar{\sigma}_k^2 + b_k^2}{c_1} + \frac{\sigma_k^2}{2c_2} \tag{91}$$

where $C = L_1 + \frac{C_1 L_4 + C_2 L_2}{\mu_1 + \mu_2} + \frac{C_1 C_2(L_3 + L_5)}{(\mu_1 + \mu_2)^2}$. Let $c_1 = c_2 = \frac{1}{2\beta}$. Then, the expectation of the difference of meta loss function is simplified.

$$\mathcal{L}\left(\theta^{k+1}\right) - \mathcal{L}\left(\theta^k\right) \leq \left(\frac{c_1 + c_2 + L_6}{2} - \frac{1}{\beta}\right)\mathbb{E}\left[\left\|\theta^{k+1} - \theta^k\right\|^2\right]$$

$$+ \frac{2C^2}{c_1}\mathbb{E}\left[\left\|\phi^*\left(\theta^k\right) - \hat{\phi}\left(\theta^k\right)\right\|\right]^2 + \frac{2\bar{\sigma}_k^2 + b_k^2}{c_1} + \frac{\sigma_k^2}{2c_2}$$

$$\leq \left(\frac{L_6}{2} - \frac{1}{2\beta}\right)\mathbb{E}\left[\left\|\theta^{k+1} - \theta^k\right\|^2\right]$$

$$+ 4\beta C^2\mathbb{E}\left[\left\|\phi^*\left(\theta^k\right) - \hat{\phi}\left(\theta^k\right)\right\|\right]^2 + 4\beta\bar{\sigma}_k^2 + 2\beta b_k^2 + \beta\sigma_k^2 \tag{92}$$

Using the lemmas we proved earlier, we present that the NashMAML algorithm always converges. Moreover, we prove that the convergent point of the NashMAML algorithm is the Stackelberg equilibrium of the stochastic optimization problem that NashMAML algorithm originally try to solve. First, let's discuss the convergence of the NashMAML algorithm.

**Theorem 3** *Let $\left(\theta^*, \phi^*\left(\theta^*\right)\right)$ be a convergent point of the NashMAML algorithm, and $\mathcal{L}\left(\theta\right)$ be an expected optimal meta loss function, that is, $\mathcal{L}\left(\theta\right) = \mathbb{E}_{\mathcal{T}_i \sim p(\mathcal{T})}\left[F\left(\theta, \phi^*\left(\theta\right)\right)\right]$. We denote $\delta$ as the convergence criterion of the inner loop and $\bar{\delta}$ as the convergence criterion of the outer loop. That is, the inner loop is converged when $\left\|\phi^* - \hat{\phi}\right\| \leq \delta$ and the outer loop is converged when $\left\|\theta^{k+1} - \theta^k\right\| \leq \bar{\delta}$. Under Assumptions 1, 2, and 3, the following statements hold.*

- *The expected difference of the meta-parameter $\theta$ is bounded as follows.*

$$
\mathbb{E}_{\mathcal{T}_i \sim p(\mathcal{T})} \left[ \left\| \theta^{k+1} - \theta^k \right\|^2 \right] \leq 4\beta^2 C^2 \mathbb{E} \left[ \left\| \hat{\phi} \left( \theta^k \right) - \phi^* \left( \theta^k \right) \right\| \right]^2
$$
$$
+ \beta^2 \sigma_k^2 + 2\beta^2 b_k^2 + 4\beta^2 \left( \bar{C}_1 + \frac{C_1 C_2}{\mu_1 + \mu_2} \right)^2
$$
$$
\leq 4\beta^2 C^2 \delta^2 + \beta^2 \sigma_k^2 + 2\beta^2 b_k^2 + 4\beta^2 \left( \bar{C}_1 + \frac{C_1 C_2}{\mu_1 + \mu_2} \right)^2
$$
(93)

- *The expected difference of the optimal meta loss function $F^* (\theta)$ is bounded as follows.*

$$
\mathcal{L} \left( \theta^{k+1} \right) - \mathcal{L} \left( \theta^k \right) \leq 4\beta^2 C^2 \left( \frac{L_6}{2} + \frac{1}{2\beta} \right) \mathbb{E} \left[ \left\| \hat{\phi} \left( \theta^k \right) - \phi^* \left( \theta^k \right) \right\| \right]^2
$$
$$
+ \left( \frac{L_6}{2} - \frac{1}{2\beta} \right) \left( \beta^2 \sigma_k^2 + 2\beta^2 b_k^2 + 4\beta^2 \left( \bar{C}_1 + \frac{C_1 C_2}{\mu_1 + \mu_2} \right)^2 \right)
$$
$$
+ 4\beta \bar{\sigma}_k^2 + 2\beta b_k^2 + \beta \sigma_k^2
$$
$$
\leq \left( \frac{L_6}{2} - \frac{1}{2\beta} \right) \left( \beta^2 \sigma_k^2 + 2\beta^2 b_k^2 + 4\beta^2 \left( \bar{C}_1 + \frac{C_1 C_2}{\mu_1 + \mu_2} \right)^2 \right)
$$
$$
+ 4\beta \bar{\sigma}_k^2 + 2\beta b_k^2 + \beta \sigma_k^2 + 4\beta^2 C^2 \delta^2 \left( \frac{L_6}{2} + \frac{1}{2\beta} \right)
$$
(94)

- *We denote the convergence speed of each non-increasing sequence $\{b_k\}_{k \geq 0}$, $\{\sigma_k\}_{k \geq 0}$, and $\{\bar{\sigma}_k\}_{k \geq 0}$, which is defined in Assumption 3, as $O(k_b)$, $O(k_\sigma)$, and $O(k_{\bar{\sigma}})$, respectively. After we choose the step size*

$$
\beta \leq \frac{\bar{\delta}}{\sqrt{4C^2 \delta^2 + 4 \left( \bar{C}_1 + \frac{C_1 C_2}{\mu_1 + \mu_2} \right)^2}}
$$
(95)

*, the iteration complexity of the NashMAML algorithm's outer loop is $O\left(\max\left\{k_b^2, k_\sigma^2, k_{\bar{\sigma}}^2\right\}\right)$ and the expected error of the optimal meta loss function of the convergent point is*

$$
\mathcal{L} \left( \theta^* \right) - \mathcal{L} \left( \theta^k \right) \leq \frac{L_6}{2} \bar{\delta}^2 + \frac{C^2 \delta^2 - \left( \bar{C}_1 + \frac{C_1 C_2}{\mu_1 + \mu_2} \right)^2}{\sqrt{C^2 \delta^2 + \left( \bar{C}_1 + \frac{C_1 C_2}{\mu_1 + \mu_2} \right)^2}} \bar{\delta}
$$
(96)

*where $C = L_1 + \frac{C_1 L_4 + C_2 L_2}{\mu_1 + \mu_2} + \frac{C_1 C_2 (L_3 + L_5)}{(\mu_1 + \mu_2)^2}$.*

*Proof.* The gradient update procedure of the meta-parameter for the NashMAML algorithm is $\theta^{k+1} = \theta^k - \beta \hat{h}_\theta \left( \theta^k, \hat{\phi} \left( \theta^k \right) \right)$. Thus, the expectation of the meta-parameter is as follows by Lemma 5.

$$
\left\| \theta^{k+1} - \theta^k \right\| = \beta \left\| \hat{h}_\theta \left( \theta^k, \hat{\phi} \left( \theta^k \right) \right) \right\|
$$
(97)

$$
\mathbb{E} \left[ \left\| \theta^{k+1} - \theta^k \right\|^2 \right] = \beta^2 \mathbb{E} \left[ \left\| \hat{h}_\theta \left( \theta^k, \hat{\phi} \left( \theta^k \right) \right) \right\|^2 \right]
$$
$$
\leq 4\beta^2 C^2 \mathbb{E} \left[ \left\| \hat{\phi} \left( \theta^k \right) - \phi^* \left( \theta^k \right) \right\| \right]^2
$$
$$
+ \beta^2 \sigma_k^2 + 2\beta^2 b_k^2 + 4\beta^2 \left( \bar{C}_1 + \frac{C_1 C_2}{\mu_1 + \mu_2} \right)^2
$$
(98)

where $C = L_1 + \frac{C_1 L_4 + C_2 L_2}{\mu_1 + \mu_2} + \frac{C_1 C_2 (L_3 + L_5)}{(\mu_1 + \mu_2)^2}$.

By Theorem 1, $\delta$-accurate estimation of the optimal joint task-specific parameter is computed with $O\left(\kappa \log \left(D / \delta\right)\right)$ number of iterations under Assumption 1.

$$\left\| \hat{\phi}\left(\theta^k\right) - \phi^*\left(\theta^k\right) \right\| \leq \delta, \forall k \tag{99}$$

Now we derive the expected difference of the optimal meta-parameter and its loss function by equations (98) and (81).

$$\begin{aligned}
\mathbb{E}\left[\left\|\theta^{k+1} - \theta^k\right\|^2\right] \leq\ & 4\beta^2 C^2 \mathbb{E}\left[\left\|\hat{\phi}\left(\theta^k\right) - \phi^*\left(\theta^k\right)\right\|\right]^2 \\
& + \beta^2 \sigma_k^2 + 2\beta^2 b_k^2 + 4\beta^2 \left(\bar{C}_1 + \frac{C_1 C_2}{\mu_1 + \mu_2}\right)^2 \\
\leq\ & 4\beta^2 C^2 \delta^2 + \beta^2 \sigma_k^2 + 2\beta^2 b_k^2 + 4\beta^2 \left(\bar{C}_1 + \frac{C_1 C_2}{\mu_1 + \mu_2}\right)^2 \tag{100}
\end{aligned}$$

$$\begin{aligned}
\mathcal{L}\left(\theta^{k+1}\right) - \mathcal{L}\left(\theta^k\right) \leq\ & \left(\frac{L_6}{2} - \frac{1}{2\beta}\right) \mathbb{E}\left[\left\|\theta^{k+1} - \theta^k\right\|^2\right] \\
& + 4\beta C^2 \mathbb{E}\left[\left\|\phi^*\left(\theta^k\right) - \hat{\phi}\left(\theta^k\right)\right\|\right]^2 + 4\beta \bar{\sigma}_k^2 + 2\beta b_k^2 + \beta \sigma_k^2 \\
\leq\ & 4\beta^2 C^2 \left(\frac{L_6}{2} + \frac{1}{2\beta}\right) \mathbb{E}\left[\left\|\hat{\phi}\left(\theta^k\right) - \phi^*\left(\theta^k\right)\right\|\right]^2 \\
& + \left(\frac{L_6}{2} - \frac{1}{2\beta}\right)\left(\beta^2 \sigma_k^2 + 2\beta^2 b_k^2 + 4\beta^2 \left(\bar{C}_1 + \frac{C_1 C_2}{\mu_1 + \mu_2}\right)^2\right) \\
& + 4\beta \bar{\sigma}_k^2 + 2\beta b_k^2 + \beta \sigma_k^2 \\
\leq\ & \left(\frac{L_6}{2} - \frac{1}{2\beta}\right)\left(\beta^2 \sigma_k^2 + 2\beta^2 b_k^2 + 4\beta^2 \left(\bar{C}_1 + \frac{C_1 C_2}{\mu_1 + \mu_2}\right)^2\right) \\
& + 4\beta \bar{\sigma}_k^2 + 2\beta b_k^2 + \beta \sigma_k^2 + 4\beta^2 C^2 \delta^2 \left(\frac{L_6}{2} + \frac{1}{2\beta}\right) \tag{101}
\end{aligned}$$

The convergence of the NashMAML algorithm is guaranteed while the meta-parameter satisfies the convergence criterion of the outer loop. Thus, the following equation holds by equation (100)

$$\begin{aligned}
4\beta^2 C^2 \delta^2 + \beta^2 \sigma_k^2 + 2\beta^2 b_k^2 + 4\beta^2 \left(\bar{C}_1 + \frac{C_1 C_2}{\mu_1 + \mu_2}\right)^2 &\leq \bar{\delta}^2 \\
\beta^2 \left(4C^2 \delta^2 + \sigma_k^2 + 2b_k^2 + 4\left(\bar{C}_1 + \frac{C_1 C_2}{\mu_1 + \mu_2}\right)^2\right) &\leq \bar{\delta}^2 \tag{102}
\end{aligned}$$

Because the convergence speed of $\sigma_k^2 + 2b_k^2$ is $O\left(\max\left\{k_b^2, k_\sigma^2\right\}\right)$, step size $\beta$ should be less than

$$\beta \leq \frac{\bar{\delta}}{\sqrt{4C^2\delta^2 + 4\left(\bar{C}_1 + \frac{C_1 C_2}{\mu_1 + \mu_2}\right)^2}} \tag{103}$$

and the difference of expected optimal meta loss $\mathcal{L}\left(\theta^{k+1}\right) - \mathcal{L}\left(\theta^k\right)$ holds by equation (101)

$$
\begin{aligned}
\mathcal{L}\left(\theta^{k+1}\right) - \mathcal{L}\left(\theta^k\right) &\leq \left(\frac{L_6}{2} - \frac{1}{2\beta}\right)\left(\beta^2\sigma_k^2 + 2\beta^2 b_k^2 + 4\beta^2\left(\bar{C}_1 + \frac{C_1 C_2}{\mu_1 + \mu_2}\right)^2\right) \\
&\quad + 4\beta\bar{\sigma}_k^2 + 2\beta b_k^2 + \beta\sigma_k^2 + 4\beta^2 C^2\delta^2\left(\frac{L_6}{2} + \frac{1}{2\beta}\right) \\
&\leq 4\beta^2\left(\frac{L_6}{2} - \frac{1}{2\beta}\right)\left(\bar{C}_1 + \frac{C_1 C_2}{\mu_1 + \mu_2}\right)^2 + 4\beta^2 C^2\delta^2\left(\frac{L_6}{2} + \frac{1}{2\beta}\right) \\
&\leq \frac{L_6\bar{\delta}^2}{2} - \frac{\bar{\delta}^2}{2\beta} + 4\beta C^2\delta^2 \\
&\leq \frac{L_6}{2}\bar{\delta}^2 + \frac{C^2\delta^2 - \left(\bar{C}_1 + \frac{C_1 C_2}{\mu_1+\mu_2}\right)^2}{\sqrt{C^2\delta^2 + \left(\bar{C}_1 + \frac{C_1 C_2}{\mu_1+\mu_2}\right)^2}}\bar{\delta}
\end{aligned}
\tag{104}
$$

at the convergent point $(\theta^*, \phi^*(\theta^*))$ with convergent speed $O\left(\max\left\{k_b^2, k_\sigma^2, k_{\bar{\sigma}}^2\right\}\right)$. That is, the error of the expected meta loss at the convergent point is less than

$$
\frac{L_6}{2}\bar{\delta}^2 + \frac{C^2\delta^2 - \left(\bar{C}_1 + \frac{C_1 C_2}{\mu_1+\mu_2}\right)^2}{\sqrt{C^2\delta^2 + \left(\bar{C}_1 + \frac{C_1 C_2}{\mu_1+\mu_2}\right)^2}}\bar{\delta}
\tag{105}
$$

Next, we discuss the convergent point of the NashMAML algorithm is the subStackelberg equilibrium of the target stochastic optimization problem of the NashMAML algorithm described in equations (8) and (9).

**Lemma 7** *The optimal expected meta loss function $\mathcal{L}(\theta)$ is always equal to the task $i$'s expected loss function $\mathbb{E}\left[\mathcal{L}_i(\theta)\right]$.*

*Proof.* Because the sampling task is done through replacement sampling, the following holds.

$$
\begin{aligned}
\mathcal{L}(\theta) &= \mathbb{E}_{\mathcal{T}_i \sim p(\mathcal{T})}\left[F\left(\theta, \phi^*(\theta)\right)\right] \\
&= \frac{1}{N}\sum_{i=1}^{N}\mathbb{E}_{\mathcal{T}_i \sim p(\mathcal{T})}\left[\mathcal{L}_i\left(\theta, \phi^*(\theta)\right)\right] \\
&= \mathbb{E}_{\mathcal{T}_i \sim p(\mathcal{T})}\left[\mathcal{L}_i\left(\theta, \phi^*(\theta)\right)\right]
\end{aligned}
\tag{106}
$$

**Theorem 4** *Let $(\theta^*, \phi^*(\theta^*))$ be an optimal solution of the following stochastic optimization problem which is the target problem of the NashMAML algorithm.*

$$
\theta^* = \arg\min_{\theta \in \mathbb{R}^d}\mathbb{E}_{\mathcal{T}_i \sim p(\mathcal{T})}\left[\mathcal{L}_i\left(\theta, \phi_i^*(\theta)\right)\right]
\tag{107}
$$

$$
\phi_i^*(\theta) = \arg\min_{\phi_i \in \Omega_i\left(\phi_{-i}^*(\theta), \theta\right)} f_i\left(\phi_i, \phi_{-i}^*(\theta), \theta\right)
\tag{108}
$$

*We denote the expected meta loss function of the stochastic optimization problem $\mathbb{E}_{\mathcal{T}_i \sim p(\mathcal{T})}\left[\mathcal{L}_i\left(\theta, \phi^*(\theta)\right)\right]$ as $\mathbb{E}\left[\mathcal{L}_i^*(\theta)\right]$. Let $\delta$ and $\bar{\delta}$ are the convergence criterion of the inner loop and the outer loop, respectively. Then, under Assumptions 1, 2, and 3, the NashMAML algorithm with step size $\beta \leq \frac{\bar{\delta}}{\sqrt{4C^2\delta^2 + 4\left(\bar{C}_1 + \frac{C_1 C_2}{\mu_1+\mu_2}\right)^2}}$ compute the optimal solution of the stochastic optimization problem described in equations (107) and (108) with the convergence speed $O\left(\max\left\{k_b^2, k_\sigma^2, k_{\bar{\sigma}}^2\right\}\right)$ and error*

$$
\mathbb{E}\left[\mathcal{L}_i^*(\theta^*)\right] - \mathbb{E}\left[\mathcal{L}_i^*(\theta^k)\right] \leq \frac{L_6}{2}\bar{\delta}^2 + \frac{C^2\delta^2 - \left(\bar{C}_1 + \frac{C_1 C_2}{\mu_1+\mu_2}\right)^2}{\sqrt{C^2\delta^2 + \left(\bar{C}_1 + \frac{C_1 C_2}{\mu_1+\mu_2}\right)^2}}\bar{\delta}
\tag{109}
$$

where $C = L_1 + \frac{C_1 L_4 + C_2 L_2}{\mu_1 + \mu_2} + \frac{C_1 C_2 (L_3 + L_5)}{(\mu_1 + \mu_2)^2}$. *That is, the convergent point of the NashMAML algorithm is also the generalized subStackelberg equilibrium of the stochastic generalized Stackelberg game described in equations (107) and (108).*

*Proof.* By Theorem 3 and Lemma 7, the difference of the expected meta loss function of the stochastic problem at the convergent point of the NashMAML algorithm is converged with convergent speed $O\left(\max\left\{k_b^2, k_\sigma^2, k_{\bar\sigma}^2\right\}\right)$.

$$\mathbb{E}\left[\mathcal{L}_i^*\left(\theta^{k+1}\right)\right] - \mathbb{E}\left[\mathcal{L}_i^*\left(\theta^k\right)\right] = \mathcal{L}\left(\theta^{k+1}\right) - \mathcal{L}\left(\theta^k\right)$$

$$\leq \frac{L_6}{2}\bar\delta^2 + \frac{C^2\delta^2 - \left(\bar{C}_1 + \frac{C_1 C_2}{\mu_1 + \mu_2}\right)^2}{\sqrt{C^2\delta^2 + \left(\bar{C}_1 + \frac{C_1 C_2}{\mu_1 + \mu_2}\right)^2}}\bar\delta \tag{110}$$

where $C = L_1 + \frac{C_1 L_4 + C_2 L_2}{\mu_1 + \mu_2} + \frac{C_1 C_2 (L_3 + L_5)}{(\mu_1 + \mu_2)^2}$. That is, the error of the expected meta loss function $\mathbb{E}\left[\mathcal{L}_i^*\left(\theta\right)\right]$ at the convergent point of the NashMAML algorithm is less than

$$\frac{L_6}{2}\bar\delta^2 + \frac{C^2\delta^2 - \left(\bar{C}_1 + \frac{C_1 C_2}{\mu_1 + \mu_2}\right)^2}{\sqrt{C^2\delta^2 + \left(\bar{C}_1 + \frac{C_1 C_2}{\mu_1 + \mu_2}\right)^2}}\bar\delta \tag{111}$$

Because the difference of the expected meta loss function $\mathbb{E}\left[\mathcal{L}_i^*\left(\theta\right)\right]$ is converged in equation (110), the convergent point of the NashMAML algorithm is also the optimal solution of the stochastic optimization problem described in equations (107) and (108). That is,

$$\mathbb{E}\left[\mathcal{L}_i^*\left(\theta^*\right)\right] - \mathbb{E}\left[\mathcal{L}_i^*\left(\theta^k\right)\right] \leq \frac{L_6}{2}\bar\delta^2 + \frac{C^2\delta^2 - \left(\bar{C}_1 + \frac{C_1 C_2}{\mu_1 + \mu_2}\right)^2}{\sqrt{C^2\delta^2 + \left(\bar{C}_1 + \frac{C_1 C_2}{\mu_1 + \mu_2}\right)^2}}\bar\delta \tag{112}$$

So far, we show that the NashMAML algorithm converges, and its convergent point is the generalized subStackelberg equilibrium of the stochastic generalized Stackelberg game that NashMAML originally targets. Now, we prove that the NashMAML algorithm always converges to the same point regardless of the initial meta-parameter or initial task-specific parameter, and irrespective of the order in which task-specific parameters are updated in the inner loop. We present the assumptions needed or the proof first.

**Assumption 4** *Suppose the following statements hold:*

- *The task $i$'s optimal loss function $\mathcal{L}_i\left(\theta, \phi^*\left(\theta\right)\right)$ is strongly convex on $\theta$.*

- *Let $G\left(\theta\right)$ be the $N$-player normal-form game modeling an inner loop of the NashMAML algorithm. The task $i$'s strategy set $\Omega_i$ is independent of the other task-specific parameters $\phi_{-i}$ or $G\left(\theta\right)$ has the unique generalized Nash equilibrium for all $\theta$.*

**Theorem 5** *Under Assumption 1, 2, and 3, the NashMAML algorithm converges to the same optimal solution of the stochastic optimization problem described in equations (107) and (108) regardless of the order of the task-specific parameters' gradient update in the inner loop. Moreover, the NashMAML algorithm converges to the optimal solution of the stochastic optimization problem described in equations (107) and (108) regardless of the initial meta-parameter and initial task-specific parameters under Assumption 4.*

*Proof.* Let $G\left(\theta\right)$ be the $N$-player normal-form game modeling an inner loop of the NashMAML algorithm. Because there is the unique convergent point of $G\left(\theta\right)$ by Lemma 2, the NashMAML algorithm converges to the same convergent point regardless of the order of the task-specific parameters' gradient update.

If $\Omega_i$ is independent of the other task-specific parameters $\phi_{-i}$, $G\left(\theta\right)$ is a normal-form game. That is, $G\left(\theta\right)$ has the unique Nash equilibrium under Assumption 1. Thus, there is the unique (generalized)

Nash equilibrium $\phi^*(\theta)$ of equation (108) under Assumption 4. Because the optimal task-specific loss function is convex on $\theta$, $\mathbb{E}_{\mathcal{T}_i \sim p(\mathcal{T})}[\mathcal{L}_i(\theta, \phi^*(\theta))]$ is convex on $\theta$. Therefore, there is the unique optimal solution $(\theta^*, \phi^*(\theta^*))$ of the stochastic optimization problem described in equations (107) and (108). That is, the NashMAML algorithm converges to the optimal solution of the stochastic optimization problem regardless of the initial meta-parameter and initial task-specific parameters.

