# OpenReview forum: "A Game Theoretic Approach to Meta-Learning: Nash Model-Agnostic Meta-Learning"
_ICLR.cc/2024/Conference — ICLR 2024 Conference Withdrawn Submission_

### Official Review · Reviewer_azMB · 2023-10-24

**Soundness:** 2 fair
**Presentation:** 3 good
**Contribution:** 2 fair
**Rating:** 5
**Confidence:** 4

**Summary:**

The main focus of this work is to expand upon the existing implicit-MAML framework and present a comprehensive analysis of bi-level optimization using the perspective of the Stackelberg game. The key contribution lies in imposing a constraint on the inner loop with the task distribution information, akin to trust region optimization. The effectiveness of this approach was evaluated through various standard benchmarks, such as few-shot regression and few-shot image classification.

**Strengths:**

Overall, this work is well written and the idea of exploiting the task structure seems reasonable.
However, there are some crucial confusing points in this work, which I will summarize in the following part.

**Weaknesses:**

1. About the research motivation.

Descriptions in page 1 “However, optimally minimizing the individual task losses in the inner loop may not essentially lead to minimizing the average loss in the outer-loop…”. Based on my understanding, the goal of MAML is to find the meta initialization that minimizes the average inner task losses. So will the failure of minimizing the outer loss results in the failure of the minimizing inner losses in a theoretical sense?
Also, the author mentioned previous MAML cannot well handle the worst-case loss. However, I failed to see the potential of Nash-MAML in alleviating this at least in comparison to (Collins et al., 2020).

2. About the order in the optimization.

It is reasonable to model the optimization with Stackelberg game. However, since the optimization is executed in a task batch way. I am afraid (i) the change of the task batch number affects the theoretical and empirical analysis (ii) the order of players results in completely different equilibrium. Though on page29, the author claims the order of players does not affect the convergence, the order defines different equilibrium. This requires more consideration particularly when the alternating gradient descent is executed in optimization.

3. About evaluation.

It seems the comparison baselines are not SOTA, and the meta reinforcement learning experiments are not included.

**Questions:**

See the weakness part.

---

### Official Review · Reviewer_XAaQ · 2023-10-30

**Soundness:** 3 good
**Presentation:** 2 fair
**Contribution:** 2 fair
**Rating:** 5
**Confidence:** 3

**Summary:**

This paper introduces a novel method to help the alignment of the learning objectives between the inner-loop and outer-loop in gradient based meta learning. Specifically, the concept of Stackelberg game theory framework is introduced to MAML and a method called NashMAML is proposed accordingly. Experiments reveal that the proposed method outperforms some baselines.

**Strengths:**

1) The idea of introducing Stackelberg equilibrium into the analysis of meta learning is novel.

2) The authors provide concrete theoretical analysis to support their proposed method.

**Weaknesses:**

I think this is an interesting paper but there are some issues to address before I can vote for acceptance.

1) My major concern is that the motivation for proposing the NashMAML is to align the upper level and lower level formulation when the goal of the outer-loop changes. For example, the objective of learning meta-parameters may extend beyond merely minimizing the average task loss.  However, if the authors are trying to address these problems,
they should provide corresponding metrics or environments in the experiments to support their claims, such as the Worst metric in [3].

2) Assumption 1 seems somewhat strong and directly leads to the convergence of global Stackelberg equilibrium. However, this assumption may not always exist for neural network, and also the converged point may not always be global equilibrium point.  It would be helpful if the authors could relax this assumption and see if the proposed method can converge to a local differential Stackelberg Equilibrium [1].

3) The proposed methods do not consistently outperform the baselines, such as in 5-way 1-shot FOMAML and 5-way 1-shot CAVIA. Additionally, it would be beneficial to introduce stronger and more recent MAML baselines, such as TSA-MAML [2].


4) There are some typos and unclear parts, such as in Table 2, where it should be CAVIA rather than CAVIA (32); in section 4.1, where it should be NashMAML instead of NashMALM; and in Table 1, where Nesterov's AGD is used without reference or explanation.


References:

[1] Convergence of Learning Dynamics in Stackelberg Games

[2] Task Similarity Aware Meta Learning: Theory-inspired Improvement on MAML

[3] Task-Robust Model-Agnostic Meta-Learning

**Questions:**

1) Could the proposed method be combined into more general frameworks, e.g. Prototypical Networks [4], rather than just gradient-based meta learning based frameworks? Doing so would broaden the scope of the proposed method.


References:

[4] Prototypical Networks for Few-shot Learning

---

### Official Review · Reviewer_SENj · 2023-11-04

**Soundness:** 3 good
**Presentation:** 3 good
**Contribution:** 2 fair
**Rating:** 5
**Confidence:** 3

**Summary:**

The authors propose a gradient-based meta-learning framework, introducing interdependencies among tasks through joint strategy sets and utility functions. This framework is reformulated as a generalized Stackelberg game where the meta-learner assumes the role of a leader while the base learners act as followers, and the NashMAML algorithm is presented to compute the generalized Stackelberg equilibrium. The design ensures that each base learner's actions or performance directly influence the outcomes of the other base learners within the inner loop. The approach's effectiveness is validated through experiments on sinusoidal regression and few-shot image classification tasks (mini-imagenet), demonstrating superior performance compared to prior methods in managing few-shot learning challenges.

**Strengths:**

- The Stackelberg game formulation of MAML is theoretically grounded. The practical formulation of NashMAML is quite simple and intuitive. *I have not checked the proofs in detail*

- Experiments on the mini-imagenet dataset show the effectiveness of the proposed approach.

**Weaknesses:**

- The proposed practical formulation bears a good resemblance to TAML [1], where a regularizer maintains the closeness of the adapted parameters to the meta-model. In addition to a discussion on the differences (of course, TAML does not model the regularizer as through a Stackleberg game), experimental comparison against TAML is also missing.

- “However, optimally minimizing the individual task losses in the inner-loop may not essentially lead to minimizing the average loss in the outer-loop. Furthermore, if the goal of the outer-loop changes, the current inner-loop formulation, which adapts the model to individual tasks independently, does not help learn the meta-parameter.” Supporting evidence(references or empirical results) for this statement will strengthen the motivation of the approach.

- The authors motivate the approach (in the introduction) by considering the worst-case performance. The connection between this setting and the proposed approach is not very apparent (perhaps I missed something?) and curious enough the authors end-up concluding that this will be part of the future work!

*Experimental weaknesses*
- The authors claim to compare against iMAML (page 8), but Table 2 lists only FOMAML, MAML and CAVIA. What motivated the authors to drop iMAML? This is important as iMAML too has a similar objective.
- Comparison against TR-MAML[2] is missing.
- Experiments on a single benchmark dataset are insufficient, especially, when some of the more recent works have achieved better performance than NashMAML. For eg. TAML achieves similar results! I encourage the authors to experiment with FC100 and tieredImagenet datasets as well, if not some of the more challenging cross domain few-show setups such as Metadataset and VTAB.
- How were the hyper-parameters $\lambda$ and $r$ tuned?

*Some minor issues:*
- Spelling mistakes :  inne-loop problem, NashMALM (page 5)
- Grammar issues: Which-decision maker who determines, M player - players
- Space issues: Space between arg and max in equations 2, 3 etc
- PRGD used before expanding it.

[1] Jamal, M. A., & Qi, G. J. (2019). Task agnostic meta-learning for few-shot learning. In Proceedings of the IEEE/CVF Conference on Computer Vision and Pattern Recognition (pp. 11719-11727).

[2] Liam Collins, Aryan Mokhtari, and Sanjay Shakkottai. Task-robust model-agnostic meta-learning. Advances in Neural Information Processing Systems, 33:18860–18871, 2020

**Questions:**

Please see the weaknesses

---

### Official Review · Reviewer_i9rv · 2023-11-05

**Soundness:** 2 fair
**Presentation:** 3 good
**Contribution:** 1 poor
**Rating:** 1
**Confidence:** 4

**Summary:**

This paper studies the model-agnostic meta-learning (MAML) as a bi-level optimization problem via a game theoretical view. The authors formulate MAML as a generalized Stackelberg game, propose a NashMAML algorithm to compute the generalized Stackelberg equilibrium of this model, and theoretically prove its convergence. They validate their approach on sinusoidal regression and few-shot image classification tasks.

**Strengths:**

This paper is well motivated - modelling MAML as a bi-level optimization problem and then a generalized Stackelberg equilibrium of this model.

The authors also provide rigorous, theoretical analysis for the proposed method. The proofs are given in detail.

The paper is well written.

**Weaknesses:**

I am worried about the novelty. Technically, I cannot see any difference between solving MAML and solving bi-level optimisation problems which has been well studied in the context of the Stackleberg game. Please clarify how your algorithm is different and advanced.

The current proofs looks highly rely on existing papers, including Facchinei & Pang (2003), Facchinei & Kanzow (2010), Malitsky (2015), Jo et al. (2023), as well as over 20 very strong assumptions on Lipschitz continuity, strong convexity, boundness, etc. These assumptions technically have assumed what the authors need to prove.

The experiments are only conducted on Mini-ImageNet, and the proposed method is only compared with FOMAML, MAML, and iMAML. Comparisons with more existing methods on more datasets are needed.

**Questions:**

Please address the above.